

# The sky remembers everything: Celestial amplitude, shadow and OPE in quadratic EFT of gravity

**Arpan Bhattacharyya**[⋆], **Saptaswa Ghosh**[†] **and Sounak Pal**[‡]

Indian Institute of Technology, Gandhinagar, Gujarat-382055, India

⋆ abhattacharyya@iitgn.ac.in , † saptaswaghosh@iitgn.ac.in , ‡ palsounak@iitgn.ac.in

## Abstract

In this paper, we compute the celestial amplitude arising from higher curvature corrections to Einstein gravity, incorporating phase dressing. The inclusion of such corrections leads to effective modifications of the theory's ultraviolet (UV) behaviour. In the eikonal limit, we find that, in contrast to Einstein's gravity, where the $u$ and $s$-channel contributions cancel, these contributions remain non-vanishing in the presence of higher curvature terms. We examine the analytic structure of the resulting amplitude and derive a dispersion relation for the phase-dressed eikonal amplitude in quadratic gravity. Furthermore, we investigate the celestial conformal block expansion of the Mellin-transformed conformal shadow amplitude within the framework of celestial conformal field theory (CCFT). As a consequence, we compute the corresponding operator product expansion (OPE) coefficients using the Burchnall-Chaundy expansion. In addition, we evaluate the OPE via the Euclidean OPE inversion formula across various kinematic channels and comment on its applicability and implications. Finally, we briefly explore the Carrollian amplitude associated with the corresponding quadratic EFT.

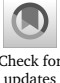

# 1  Introduction

Scattering amplitudes are among the most fundamental observables in theoretical and experimental physics. They play a central role in high-energy experiments and are of significant theoretical interest in both quantum field theory and quantum gravity. The AdS/CFT correspondence [1] provides a powerful holographic framework for understanding quantum gravity in anti-de Sitter (AdS) space. This naturally raises the question: can a similar holographic duality be formulated in flat spacetime? In flat space, the S-matrix elements serve as the primary observables. Consequently, it is natural to seek a holographic correspondence analogous to AdS/CFT in this setting. A compelling proposal in this direction is the Celestial Conformal Field Theory (CCFT), introduced in a series of works [2–6]. In later years CCFT gains much attention in the context of flat space holography, Soft theorems and asymptotic symmetries [7–12]. CCFT aims to provide a holographic description of quantum gravity in asymptotically flat spacetimes by recasting bulk quantum field theory in Minkowski space as a two-dimensional conformal field theory defined on the celestial sphere. In this framework, the Virasoro generators correspond to superrotations—local conformal transformations of the celestial sphere, which resides at null infinity where asymptotic states are defined. These superrotations extend the global Lorentz transformations, traditionally associated with the Möbius subgroup of the conformal group, to an infinite-dimensional local symmetry algebra. This enhancement of symmetry is central to celestial holography, where scattering amplitudes are reinterpreted as correlation functions of a two-dimensional CFT on the celestial sphere. Each 4d massless external scalar corresponds to a scalar primary operator in CCFT with conformal dimensions which lie in principle series i.e. $\Delta_i = 1 + i\lambda_i$ with $\lambda_i \in \mathbb{R}$.

In conventional quantum field theory, scattering amplitudes are typically computed by considering the external states as momentum eigenstates, ensuring momentum conservation at each interaction vertex. However, an alternative exists, namely the conformal primary basis. In this basis, the external states are labeled by their conformal dimensions and their positions on the celestial two-sphere, parametrized by complex coordinates $z, \bar{z}$. For massless external states, the conformal primary basis is obtained via a Mellin transformation over the energy variable $\omega$. Scattering amplitudes expressed in this basis are referred to as *celestial amplitudes*, owing to their kinematic structure and interpretation as correlation functions on the celestial sphere [13].

A comprehensive understanding of celestial conformal field theory (CCFT) necessitates knowledge of its full operator spectrum. In standard two-dimensional CFTs, the conformal block decomposition [14, 15] serves as a fundamental tool for probing the spectrum. Analogously, the block decomposition of celestial correlators and OPE has been extensively explored in [16–25]. However, unlike standard CFTs, the computation of the block expansion in CCFT differs significantly due to the kinematic constraints imposed by four-dimensional scattering

amplitudes, which result in a delta function $\delta(|z - \bar{z}|)$ in cross-ratio space. This constraint enforces the cross-ratio $z$ to be real-valued. To overcome this limitation, one can perform a shadow transformation [26–28] on one of the external operators, which relaxes the kinematic constraints and restores complex $z$ dependence in the celestial correlator [19]. This procedure enables the application of standard 2D CFT techniques to analyze the conformal block decomposition, particularly in extracting operator product expansion (OPE) coefficients. To extract the OPE data for the shadow-transformed amplitude in quadratic EFT, we employ the OPE inversion formula [29–31]. The Euclidean inversion formula, applicable to CCFT, is relatively straightforward compared to the Lorentzian version, which relies on computing the double discontinuity across branch cuts of hypergeometric functions in the complexified cross-ratio space where $z$ and $\bar{z}$ are independent variables. The key idea is to identify OPE coefficients as residues of poles in the analytically continued partial wave expansion coefficients. This analytic continuation enables the extraction of OPE data in terms of integrals over partial waves, offering a powerful method for decoding the celestial operator spectrum.

In spite of these promising ideas and newer developments in CCFT in recent years, Celestial holography has some drawbacks that need to be addressed. Most of the examples of celestial amplitude presented in the literature, especially for the UV incomplete theories, give non-analytic (or purely divergent) results [32], which is because the Mellin transformation of flat space amplitude (at some order of perturbation) integrates over full energy scale (IR to UV) of the external asymptotic states. For example, in Einstein's GR, the Celestial amplitude is purely divergent at any order of perturbation. This issue is, to some extent, resolved in the context of string theory [33–35] because of the UV softness of the theory. However, in the context of QFT, there are several approaches to handle the problem: the first one is to treat the celestial amplitude as a distribution [4]. Secondly, by regularizing it using the background field method [20, 21, 36–39]. Thirdly, by introducing a modified Mellin transformation [40]. More recently, another approach has been proposed in [41] based on the eikonal exponentiation of scattering amplitude (in the context of celestial amplitude see [37]) in both flat and curved space [42]. In the eikonal limit (small angle, high energies), the perturbative series of the scattering amplitude can be exponentiated [43, 44] and the full scattering matrix can be written in terms of the Born amplitude dressed by a phase. It has also been shown that because of the oscillating nature of the eikonal amplitudes, they are meromorphic in conformal dimensions with an infinite number of poles in the negative real axis. This provides an example of meromorphic and non-perturbative gravitational celestial amplitude. From the point of view of UV completeness, we expect improved analytic behaviour of celestial amplitude in higher derivative EFTs of gravity, which is one of the main themes of the paper. As previously noted, celestial amplitudes in general relativity (GR) exhibit non-analytic—more precisely, *distributional*— behaviour. In this work, we aim to explore whether this singular structure persists in higher-derivative EFTs, specifically within the context of *quadratic gravity*. Our investigation centres around the following key questions:

- Does the celestial amplitude in quadratic gravity remain distributional, or does it exhibit improved analytic properties?

- Is the introduction of *eikonal resummation* necessary in such theories to achieve better analytic behaviour in the celestial amplitude?

- Can we get some analytic control to extract the OPE data (including spinning exchanges) from the celestial correlator corresponding to the Born amplitude.

- Furthermore, is it possible to extract OPE data from the *non-perturbative (in couplings of the theory) eikonal amplitude*?

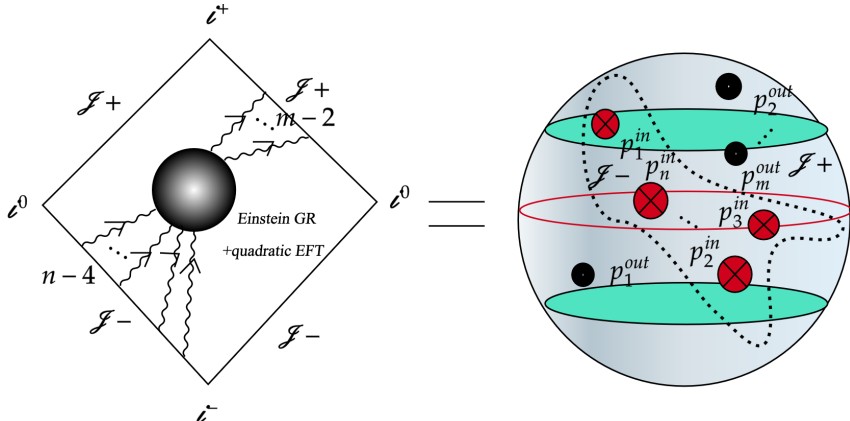

Figure 1: Figure describing the celestial description of conformal correlator (right) for $n \to m$ scattering amplitude in asymptotically flat space (left).

Through these questions, we aim to understand the interplay between analytic structure, resummation techniques, and CFT data for the celestial amplitudes of massless scalar in higher-derivative theories.

This paper is organised as follows. In Section (2), we briefly review the key theme of the paper, i.e. eikonal amplitude in the celestial sphere. Apart from this, we also discuss the boost eigenstate and its connection to the Mellin transform. In Section (3), we discuss the calculation of the amplitude in terms of Mandelstam invariants and then proceed to generalise it for quadratic EFT. In this section, we also describe analyticity and dispersion relation(s) relevant to our context, generalising from pure GR scenario. The nature of the $\gamma$-poles in the celestial amplitude for quadratic EFT is discussed. Further in Section (4), we discuss Celestial *Operator Product Expansion (OPE)* and show the detailed computation of the conformal block decomposition in Celestial conformal field theory (CCFT). To compute the Euclidean OPE coefficients, we used the *OPE-inversion formula* pioneered by Caron-Huot. In Section (5), we calculate the carrollian amplitude from the celestial counterpart and calculate the corrections we obtain from the non-vanishing quadratic EFT coefficients. We also demonstrate how the IR pole in carrollian amplitude shifts its value from that of GR. In Appendix A, we show the derivation and the simplification of the Mellin integration we encountered in terms of the *simplex variables*.

## 2 Celestial amplitude: A lightning review

Traditionally, the external states of scattering amplitudes are parametrized by the energy eigenvalues. Alternatively, the conformal primary basis is a particularly interesting alternative, which renders the resulting scattering amplitude in a form that transforms it into a conformal correlator on the celestial sphere schematically shown in Fig. (1). As such, scattering amplitudes expressed in the conformal primary basis are usually referred to as celestial amplitudes. For the massless field scattering, external states in the four-momentum basis can be parametrized in terms of frequency $\omega$ and a point on celestial sphere $(z, \bar{z})$ as the external momenta can be parametrized as [13, 32]

$$p_\mu^i = \frac{\omega}{\sqrt{2}} \left( 1 + |z_i|^2, z_i + \bar{z}_i, -i(z_i - \bar{z}_i), 1 - |z_i|^2 \right). \tag{1}$$

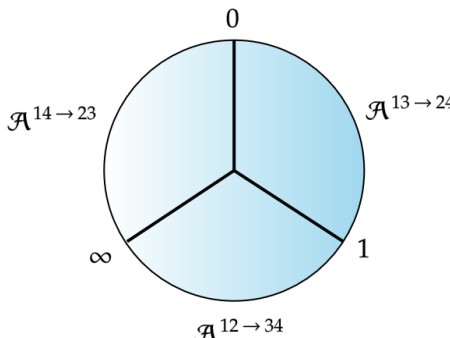

Figure 2: Figure depicting the regime of validity for Mellin transformed amplitude(s) in different kinematical regime.

In celestial conformal field theory (CCFT), the massless four-momenta can be written as (1). In $(-+++)$ convention $z$ is the cross-ratio and $\bar{z}$ is its complex conjugate,

$$z = \frac{z_{12}z_{34}}{z_{13}z_{24}}, \qquad \bar{z} = \frac{\bar{z}_{12}\bar{z}_{34}}{\bar{z}_{13}\bar{z}_{24}}. \tag{2}$$

In CCFT, $\mathcal{O}_{\Delta, m=0}$ is a conformal primary with conformal dimension,

$$h_i = \frac{\Delta_i + J_i}{2}, \qquad \bar{h}_i = \frac{\Delta_i - J_i}{2}, \qquad \text{with } \Delta_i = 1 + i\lambda_i, \lambda_i \in \mathbb{R}. \tag{}$$

The 4d scattering channels are given as

$$
\begin{aligned}
a) & \quad 12 \Longleftrightarrow 34 & (z > 1), \\
b) & \quad 13 \Longleftrightarrow 24 & (0 < z < 1), \\
c) & \quad 14 \Longleftrightarrow 23 & (z < 0).
\end{aligned} \tag{3}
$$

Now, in these three kinematic regimes, the corresponding celestial amplitude, which is a Mellin transform of flat space scattering amplitude, transforms like a $n$-point correlator (for our case, we took the primaries to be scalars) of a two-dimensional conformal field theory and is given by

$$\mathcal{A}_n(z_i, \bar{z}_i) := \left\langle \prod_{i=1}^n \mathcal{O}_{\Delta_i}(z_i, \bar{z}_i) \right\rangle = \int_0^\infty \prod_{i=1}^n d\omega_i \, \omega_i^{\Delta_i - 1} \mathbb{M}_n(p_1, \ldots, p_n) \delta^{(4)}\left( \sum_{i=1}^n \epsilon_i \omega_i q_i \right). \tag{4}$$

In this paper, we will focus on the 4-point scalar amplitude, which is given by (using the $\text{SL}(2, \mathbb{C})$ invariance)

$$\mathcal{A}_{\Delta_i, J_i}(z_i, \bar{z}_i) = \frac{\left(\frac{z_{14}}{z_{13}}\right)^{h_3 - h_4} \left(\frac{z_{24}}{z_{14}}\right)^{h_1 - h_2} \left(\frac{\bar{z}_{14}}{\bar{z}_{13}}\right)^{\bar{h}_3 - \bar{h}_4} \left(\frac{\bar{z}_{24}}{\bar{z}_{14}}\right)^{\bar{h}_1 - \bar{h}_2}}{z_{12}^{h_1 + h_2} z_{34}^{h_3 + h_4} \bar{z}_{12}^{\bar{h}_1 + \bar{h}_2} \bar{z}_{34}^{\bar{h}_3 + \bar{h}_4}} \mathcal{G}_{\Delta_i, J_i}(z_i, \bar{z}_i), \tag{5}$$

where the left and right moving conformal dimensions are

$$h_i + \bar{h}_i = \Delta_i, \qquad h_i - \bar{h}_i = J_i. \tag{6}$$

Using the $\text{SL}(2, \mathbb{C})$ invariance we can fix the coordinates $z_i$ to

$$z_1 = 0, \qquad z_2 = z, \qquad z_3 = 1, \qquad z_4 = \infty. \tag{7}$$

Table 1: The physical regions, C.O.M energy $\omega$ and Mandelstam variables in the three different kinematic regimes.

| Kinematics | $12 \leftrightarrow 34$ | $13 \leftrightarrow 24$ | $14 \leftrightarrow 23$ |
|---|---|---|---|
| physical region | $z \geq 1$ <br> $s \geq 0 \geq u,t$ | $1 \geq z \geq 0$ <br> $u \geq 0 \geq s,t$ | $0 \geq z$ <br> $t \geq 0 \geq s,u$ |
| $(s,u,t)$ | $(\omega^2, -\frac{1}{z}\omega^2, -\frac{(z-1)}{z}\omega^2)$ | $(-z\omega^2, \omega^2, -(1-z)\omega^2)$ | $(-\frac{(-z)}{1-z}\omega^2, -\frac{1}{1-z}\omega^2, \omega^2)$ |

Now the momentum conservation further reduces the amplitude to be[1]

$$\mathcal{G}_{\Delta_i, J_i}(z_i, \bar{z}_i) = (z-1)^{\frac{\Delta_1 - \Delta_2 - \Delta_3 + \Delta_4}{2}} \delta(iz - i\bar{z}) \mathcal{A}(\boldsymbol{\Delta}, J_i, z), \tag{9}$$

where $\boldsymbol{\Delta} = \sum_i \Delta_i$ and the delta function ensures the planarity of the celestial amplitude. In the celestial sphere, we have different kinematic channels depending on the values of $\epsilon_i$.

The theory dependent dynamics of the scattering amplitude is encoded in $\mathcal{A}(\gamma, z)$, given by

$$\mathcal{A}(\gamma, z) \to \mathcal{A}^{ij \to kl}(\gamma, z) = B^{ij \to kl}(z) \int_0^\infty d\omega\, \omega^{\gamma - 1} \mathbb{M}\left(s = \Phi(z)\omega^2, t = -\phi(z)\omega^2\right), \tag{10}$$

where $\gamma$ is related to $\boldsymbol{\Delta}$ as, $\gamma = \sum_{i=1}^4 (\Delta_i - 1) = \boldsymbol{\Delta} - 4$. Also, the functions $\Phi(z)$ and $\phi(z)$ are given in Table (1) for different kinematic regimes. Now, we proceed to the central theme of the paper. We investigate the eikonal limit of the celestial amplitude for quadratic EFT.

## 3 Celestial eikonal amplitude for quadratic gravity

We consider scattering amplitudes with external massless scalars, with the exchange particle being a graviton which has one massive mode due to non-vanshing coupling constants of quadratic EFT. The main ingredient for computing the celestial eikonal amplitude is the Born amplitude in quadratic gravity. The gravity theory we are going to consider is the following,

$$S_g = \int d^4x \sqrt{-g}\left(\kappa \mathcal{R} + \alpha \mathcal{R}_{\mu\nu}\mathcal{R}^{\mu\nu} - \frac{1}{3}(\alpha + \beta)\mathcal{R}^2\right), \tag{11}$$

where, $\kappa \sim m_p^2 \sim \frac{1}{G}$ and $\alpha, \beta$ are dimensionless Wilson coefficients (or coupling constants of the theory) of the EFT. At the tree level, we have the following Feynman diagram(s) as depicted in Fig (3). To compute the diagram, we need to know the graviton propagator, which is given by [45]

$$\kappa \langle h_{\mu\nu}(q) h_{\eta\delta}(-q)\rangle = 2\mathcal{P}^{(2)}_{\mu\nu;\eta\delta}\left(\frac{1}{q^2} - \frac{1}{q^2 + \frac{\kappa}{\alpha}}\right) + \mathcal{P}^{(0)}_{\mu\nu;\eta\delta}\left(-\frac{1}{q^2} + \frac{1}{q^2 + \frac{\kappa}{2\beta}}\right), \tag{12}$$

where the projectors are defined as

$$\mathcal{P}^{(0)}_{\mu\nu;\eta\delta} = -\frac{1}{3q^2}\left(q_\mu q_\nu \eta_{\eta\delta} + q_\eta q_\delta \eta_{\mu\nu}\right) + \frac{1}{3}\eta_{\mu\nu}\eta_{\eta\delta} + \frac{1}{3q^4}q_\mu q_\nu q_\eta q_\delta, \tag{13}$$

---

[1]The momentum conserving delta function can also be written in terms of the simplex variables [4] (we describe it in detail in Appendix A) $\sigma_i = v^{-1}\omega_i$ with $\sum_{i=1}^n \sigma_i = 1$,

$$\prod_{i=1}^n \int_0^\infty d\omega_i\, \omega_i^{i\lambda_i}[\cdots] = \int_0^\infty dv\, v^{-1 + \sum i\lambda_i} \prod_{i=1}^n \int_0^1 d\sigma_i\, \sigma_i^{i\lambda_i} \delta\left(\sum_i \sigma_i - 1\right)[\cdots]. \tag{8}$$

For a detailed derivation of the $\sigma$ integral and delta function simplification, we refer the reader to look at Appendix (A).

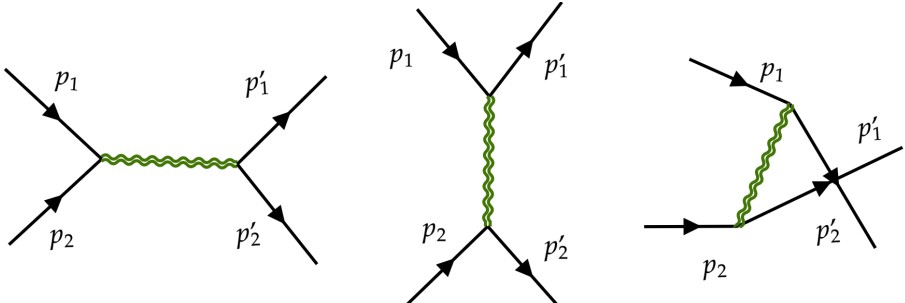

Figure 3: Figure describing the $s, t, u$-channel diagram respectively, relevant for our computation.

and

$$
\begin{aligned}
\mathcal{P}^{(2)}_{\mu\nu;\eta\delta} &= \frac{1}{3q^2}\left(q_\mu q_\nu \eta_{\eta\delta} + q_\eta q_\delta \eta_{\mu\nu}\right) - \frac{1}{2q^2}\left(q_\mu q_\eta \eta_{\nu\delta} + q_\mu q_\delta \eta_{\nu\eta} + q_\nu q_\eta \eta_{\mu\delta} + q_\nu q_\delta \eta_{\mu\eta}\right) \\
&+ \frac{2}{3q^4}\left(q_\mu q_\nu q_\eta q_\delta\right) + \frac{1}{2}\left(\eta_{\mu\eta}\eta_{\nu\delta} + \eta_{\mu\delta}\eta_{\nu\eta}\right) - \frac{1}{3}\eta_{\mu\nu}\eta_{\eta\delta} .
\end{aligned}
\tag{14}
$$

Before proceeding to the main discussion, we begin by outlining several foundational aspects of quadratic effective field theories (EFTs) of gravity. It is well understood that quantum corrections to Einstein gravity generically induce higher-derivative terms, including those quadratic in the curvature, such as $R^2$. This naturally motivates the study of such terms at the level of the microscopic (bare) action, providing a framework amenable to consistent perturbative renormalization. The inclusion of curvature-squared contributions in classical gravity was first proposed in [46], and their role in rendering gravity power-counting renormalizable was subsequently demonstrated in [47], with full renormalizability to all orders in perturbation theory established in [48].

Despite these notable achievements, higher-derivative gravitational theories are often regarded as problematic due to issues related to unitarity. This concern becomes manifest at the level of the tree-level propagator (12), where the ultrablack behavior improves from a $1/q^2$ to a $1/q^4$ fall-off, yet this improvement introduces a massive spin-2 ghost with mass $m = \sqrt{\kappa/\alpha}$, signaling a breakdown of perturbative unitarity. More specifically, while it is possible to quantize the theory such that all excitations have positive-definite energy, this necessitates the inclusion of negative-norm (ghost) states in the Hilbert space. As shown in [49], such states inevitably violate unitarity and cannot be consistently projected out without sacrificing the unitarity of the $S$-matrix. Therefore, despite the theory's favorable UV behavior, its physical consistency remains fuzzy in the absence of a *precise* resolution to the unitarity problem (see also the Appendix of [48] for a detailed discussion).

However, as emphasized in [50], unitarity is fundamentally a dynamical property, and cannot be definitively assessed within the confines of a purely perturbative treatment or at the level of the tree-level propagator alone. A comprehensive analysis requires accounting for loop corrections and potentially non-perturbative effects at fixed energy. As we will demonstrate in the following sections, the eikonal scattering amplitude constructed within a bottom-up approach—non-perturbative in the coupling—is manifestly unitary by construction.

Nevertheless, more recent analyses [51–54] have shown that Einstein gravity, when treated as a quantum theory, violates tree-level unitarity in the high-energy limit, failing to satisfy the unitarity bound $|\mathbb{A}(s,t)| < C$, where $C$ is a finite constant. In particular, in the Regge limit, the amplitude in general relativity exhibits unbounded growth, scaling as $|\mathbb{A}(s,t)|_{\mathrm{GR}} \sim O(s^1) \to \infty$. In contrast, the same studies demonstrate that scalar matter scat-

tering mediated by graviton exchange in quadratic gravity satisfies unitarity bound at high energies, despite the presence of negative-norm (ghost) states. This result is in accord with the expectations of the Llewellyn Smith conjecture [55], which heuristically suggests that renormalizable quantum field theories should also exhibit unitary behavior.

In addition, our subsequent computations of the tree-level scattering amplitude in quadratic gravity yields the following structure:

$$\mathbb{A}(s,t) \sim \frac{s^2}{t} + O(s^0), \tag{15}$$

where the leading term arises from the Einstein-Hilbert (GR) sector, while the subleading constant contribution originates from the quadratic curvature corrections in the effective action. When the theory is treated as an effective field theory, the eikonal limit $t/s \to 0$ remains well-defined by taking the momentum transfer $t$ to be sufficiently small and keeping the center-of-mass energy $s$ bounded by the EFT cutoff $\Lambda$. Moreover, it is straightforward to observe that the quadratic curvature contributions exhibit improved high-energy behavior compared to the GR term. Consequently, there is no apparent violation of the unitarity bound—at least within the regime where the theory is interpreted as a low-energy effective description of some UV-complete quantum gravity theory.

Under this interpretation, Einstein gravity treated as an effective field theory (EFT) remains consistent with unitarity, as the amplitude stays within unitarity bound. Importantly, the constant contribution from the higher-derivative terms does not introduce any violation of the unitarity bound; that is, one finds

$$|\mathbb{A}(s,t)| \leq f(\Lambda) + O(s^0), \tag{16}$$

for some finite increasing function $f(\Lambda)$. In summary, when quadratic gravity (or Einstein gravity) is treated as a low-energy EFT, unitarity of the scattering amplitude is not much problematic (rather, in another context, i.e, in the S-matrix bootstrap program, the goal is to put a bound on the EFT coupling, assuming the existence of a high-energy UV completion which is causal and unitary). This contrasts with the situation where one attempts to promote the theory to a full UV-complete quantum field theory, where issues such as ghost modes and associated violations of unitarity may become significant.

Keeping these takeaways in mind, we now set up the scattering process for our case. Specifically, we consider the gravitationally mediated scattering of massless scalar matter fields. The dynamics of the scalar field are governed by the following matter action:

$$S_m = \int d^4x \sqrt{-g} \left( -\frac{1}{2} g^{\mu\nu} \nabla_\mu \phi \nabla_\nu \phi \right), \tag{17}$$

where $\phi$ denotes a massless scalar field minimally coupled to gravity and bulk gravitational action is defined in (11). Now to compute the amplitude in terms of the Mandelstam variables $s, t, u$ we use the constraint $s + t + u = 0$ for external massless on-shell states. Then the amplitude $\mathbb{M}(s,t)$ can be cast as the sum of three distinct channels $(s,t,u)$,[2] after which we take the *eikonal limit,* $\frac{t}{s} \to 0$, to extract the dominant contribution. It is noteworthy that, similar to GR without higher curvature corrections, the dominant eikonal contribution comes from the $t$ channel. The key difference is the contribution from $s$ and $u$ channels, which cancel

---

[2]The Mandelstam variables are given by

$$s := -(p_1 + p_2)^2, \qquad t := -(p_1 - p_3)^2, \qquad u := -(p_1 - p_4)^2. \tag{18}$$

each other in GR but not if we have higher curvature terms. We can promptly write down the contributions to the amplitude from the three channels as

$$\mathbb{M}^{\text{EFT}}_{\text{Born}}(s,t) = \mathbb{A}_s(s,t) + \mathbb{A}_t(s,t) + \mathbb{A}_u(s,t), \tag{19}$$

where

$$\mathbb{A}_s(s,t) = \frac{2}{24}\left(s^2 + 6st + 6t^2\right)\left(\frac{1}{-s} - \frac{1}{\frac{\kappa}{\alpha} - s}\right) - \frac{1}{48}s^2\left(\frac{1}{-s} - \frac{1}{\frac{\kappa}{2\beta} - s}\right),$$

$$\mathbb{A}_t(s,t) = \frac{2}{24}\left(6s^2 + 6st + t^2\right)\left(\frac{1}{-t} - \frac{1}{\frac{\kappa}{\alpha} - t}\right) - \frac{1}{48}t^2\left(\frac{1}{-t} - \frac{1}{\frac{\kappa}{2\beta} - t}\right), \tag{20}$$

$$\mathbb{A}_u(s,t) = \frac{2}{24}\left(s^2 - 4st + t^2\right)\left(-\frac{1}{\frac{\kappa}{\alpha} + (s+t)} - \frac{1}{-s-t}\right) - \frac{1}{48}(s+t)^2\left(-\frac{1}{\frac{\kappa}{2\beta} + (s+t)} - \frac{1}{-s-t}\right).$$

After adding all of them, in the eikonal limit,[3]

$$\mathbb{M}^{\text{EFT}}_{\text{Born}}(s,t) \to -\frac{1}{48}s\left(\frac{(3\kappa + \alpha s - 8\beta s)}{(\kappa - \alpha s)(\kappa - 2\beta s)} - \frac{4}{\kappa + \alpha s} + \frac{1}{\kappa + 2\beta s} + \frac{24(-s)}{-t\kappa}\right). \tag{21}$$

In the limit $\alpha \to 0, \beta \to 0$, it gives back the Einstein-GR result, $\mathbb{M}^{\text{GR}} \to \frac{s^2}{-2t}$. For non-zero values of $\alpha$ and $\beta$, we get the tree-level amplitude of the quadratic EFT. This can be written in the following way,

$$\mathbb{M}^{\text{EFT}}_{\text{Born}}\left(s = \omega^2, t = -\frac{z-1}{z}\omega^2\right)$$
$$= -\frac{1}{48}\omega^2\left(\frac{3\kappa + \alpha\omega^2 - 8\beta\omega^2}{(\kappa - \alpha\omega^2)(\kappa - 2\beta\omega^2)} - \frac{4}{\kappa + \alpha\omega^2} + \frac{1}{\kappa + 2\beta\omega^2} - \frac{24z}{(z-1)\kappa}\right). \tag{22}$$

The $z$ should be chosen properly for different kinematic regions as given in Table (1). One should note that even though we get a non-zero finite piece due to the presence of non vanishing $\alpha, \beta$, in the $s$ and $u$-channel, the $t-$channel contribution in the eikonal limit remains the same as of GR.

As discussed previously there are three distinct kinematic regions: $\mathbf{12 \to 34}$, $\mathbf{13 \to 24}$, and $\mathbf{14 \to 23}$. These regions differ significantly from one another and are each valid only within specific ranges of the cross-ratios. To go from one kinematical region to another, one typically employs a systematic procedure of analytic continuation. This process is illustrated schematically in Fig. (2). Now, the celestial amplitude (for external scalar operators) in three different kinematical regions is given by

$$\mathcal{A}^{12\to34}(\Delta, z) = 2^{3-\gamma}z^2\int_0^\infty d\omega\,\omega^{\gamma-1}\mathbb{M}^{\text{EFT}}_{\text{Born}}\left(s = \omega^2, t = -\frac{z-1}{z}\omega^2\right) \qquad (z>1)$$

$$= \frac{i\pi 2^{-\frac{3\gamma}{2}-2}z^2\left(\alpha^{\frac{\gamma}{2}+1} - 2^{\frac{\gamma}{2}+3}\beta^{\frac{\gamma}{2}+1}\right)\left(\frac{\alpha\beta}{\kappa}\right)^{-\frac{\gamma}{2}-1}}{3\left(-1+e^{\frac{i\pi\gamma}{2}}\right)\kappa} + \underbrace{\frac{2^{3-\gamma}\pi z^3\delta(i(\gamma+2))}{(z-1)\kappa}}_{\text{Einstein gravity}}$$

$$= \delta_1(\alpha,\beta|\gamma)z^2 + \delta_2(\gamma)\frac{z^3}{(z-1)\kappa},$$

---

[3]We use the notation $\mathbb{M}^{\text{EFT}}_{\text{Born}}$ to denote the Born amplitude in the eikonal limit, as this approximation is employed throughout the manuscript in place of the full Born amplitude. Subsequently, we take the Mellin transformation of the eikonal limit of the Born amplitude to find the celestial amplitude.

$$\mathcal{A}^{13 \to 24}(\Delta, z) = 2^{3-\gamma} z^{2+\frac{\gamma}{2}} \int_0^\infty d\omega\, \omega^{\gamma-1}\, \mathbb{M}^{\text{EFT}}_{\text{Born}}\left(s = -z\omega^2, t = -(1-z)\omega^2\right) \qquad (0 < z < 1)$$

$$\to \mathcal{A}^{12 \to 34}(\Delta, z),$$

$$\mathcal{A}^{14 \to 23}(\Delta, z) = 2^{3-\gamma}(-z)^{2+\frac{\gamma}{2}}(1-z)^{-\frac{\gamma}{2}} \int_0^\infty d\omega\, \omega^{\gamma-1}\, \mathbb{M}^{\text{EFT}}_{\text{Born}}\left(s = \frac{z}{1-z}\omega^2, t = \omega^2\right) \qquad (z < 0)$$

$$\to \mathcal{A}^{12 \to 34}(\Delta, z), \tag{23}$$

where

$$\delta_1(\alpha, \beta | \gamma) = \pi\, e^{-\frac{i\pi\gamma}{4}} \frac{2^{-\frac{3\gamma}{2}-2} z^2 \kappa^{\gamma/2} \left(\beta^{-\frac{\gamma}{2}-1} - 2^{\frac{\gamma}{2}+3} \alpha^{-\frac{\gamma}{2}-1}\right)}{6 \sin\left(\frac{\pi\gamma}{4}\right)}, \qquad \delta_2(\gamma) = 2^{3-\gamma} \pi \delta(i(\gamma+2)). \tag{24}$$

Collecting the result of the integrals, we can write the full amplitude in the conformal basis, as

$$\mathcal{A}_4(z, \bar{z}) \sim (z-1)^{\frac{\Delta_1 - \Delta_2 - \Delta_3 + \Delta_4}{2}} |z|^{-\Delta_1 - \Delta_2} \frac{\delta(|z - \bar{z}|)}{|z_{13}|^2 |z_{24}|^2} \mathcal{A}^{ij \to kl}(\Delta, J_i, z). \tag{25}$$

We find that the Mellin-transformed Born amplitude has a better analytic structure than GR, which is purely divergent (or purely distributional), which answers the first question we rise in the introduction. Moreover, we find that the part coming from the quadratic correction in the amplitude is analytic in the whole complex $\gamma$-plane except $\gamma = n$, where $n \in \mathbb{Z}$. In contrast, for GR the celestial amplitude in the eikonal limit is non-analytic, which replicates the fact that quadratic EFT modifies the analytic behaviour in the whole energy plane, and correspondingly, the amplitude becomes analytic in the $\gamma$ plane with the location of isolated singularities.[4] Next, we proceed to discuss the eikonal amplitude in GR and then generalize it for quadratic EFT.

## 3.1 Eikonal amplitude in GR

The eikonal approximation provides an effective framework for analyzing high-energy scattering processes in the regime where momentum transfer is small compared to the centre-of-mass energy. In gravity, this leads to an eikonal amplitude dominated by ladder-type graviton exchanges, capturing the resummation of leading contributions at each order in perturbation theory [43, 44]. Remarkably, the eikonal amplitude encodes both ultrablack (UV) and infrared (IR) aspects of the theory. The eikonal regime thus serves as a stage for exploring how quantum gravity reconciles high-energy scattering with universal IR behaviour governed by asymptotic symmetries.

Before going into the eikonal phase calculations of quadratic higher curvature gravity, we briefly describe the computation in the context of GR. The eikonal expression for the amplitude is given by summing the eikonal phase. We take bottom-up approach to compute the eikonal amplitude. In bottom-up approach we define the eikonal phase as [44, 56, 57]

$$\chi = \frac{2\pi G}{E p} \int \frac{d^{D-2} \boldsymbol{q}_\perp}{(2\pi)^{D-2}} e^{i\boldsymbol{q}_\perp \cdot \boldsymbol{x}}\, \mathbb{M}^{\text{EFT}}_{\text{Born}}\left(s, t = -\boldsymbol{q}_\perp^2\right). \tag{26}$$

Consequently the eikonal amplitude is given by

$$\mathcal{M}_{\text{eik}}(s, t) = 8E p \int d^{D-2} \boldsymbol{x}_\perp e^{-i\boldsymbol{q}_\perp \cdot \boldsymbol{x}_\perp} (e^{i\chi} - 1). \tag{27}$$

---

[4]A general proof in [32] shows that for QFT's with better UV behaviour(s) leads to analytic amplitudes (rather than purely divergent distributional nature) with poles in the right-half (left-half) complex $\gamma$-plane which we call 'UV poles' (IR poles).

In GR the eikonal amplitude is given by [43, 44]

$$i\mathcal{M}_{\text{eik}}^{\text{GR}} = \frac{8\pi Ep}{\mu^2} \frac{\Gamma\left(1 - \frac{iG\frac{s^2}{2}}{2Ep}\right)}{\Gamma\left(\frac{iG\frac{s^2}{2}}{2Ep}\right)} \left(\frac{4\mu^2}{q_\perp^2}\right)^{1 - \frac{iG\frac{s^2}{2}}{2Ep}} = - \underbrace{\frac{16i\pi Gs^2}{2t}}_{\text{Born Amplitude}} \times \underbrace{\frac{\Gamma\left(-\frac{iGs^2}{4Ep}\right)}{\Gamma\left(\frac{iG\frac{s^2}{2}}{2Ep}\right)} \left(\frac{4\mu^2}{q_\perp^2}\right)^{-\frac{iGs^2}{4Ep}}}_{\text{Phase}}. \tag{28}$$

This is an important feature that the eikonal (leading) amplitude in GR to all order of $G$ is the product of the born amplitude and a phase factor.

## 3.2 Eikonal amplitude in quadratic gravity

Now for quadratic EFT of gravity, as the propagator changes, the expression of eikonal phase $\chi$ also changes. A natural question that arises in the context of high-energy gravitational scattering is whether eikonal exponentiation remains valid in higher-derivative theories such as quadratic gravity. The answer, we argue, is affirmative in the eikonal regime, and we provide the reasoning below. The validity of eikonal exponentiation in quadratic gravity can be addressed by considering the structure of gravitational scattering amplitudes in the high-energy, small-angle (eikonal) regime. In this limit, the dominant contributions arise from the exchange of soft, long-wavelength gravitons. Crucially, this infrared (IR) behavior is governed by the long-distance propagation of the gravitational field, which remains dictated by the Einstein-Hilbert term—even in the presence of higher-curvature corrections such as $R^2$, $R^2_{\mu\nu}$, or $R^3$. These UV modifications primarily affect the short-distance (non-eikonal) part of the interaction and do not interfere with the leading IR dynamics responsible for exponentiation.

Within the effective field theory framework, and under the assumption that the higher-derivative couplings (e.g., $\alpha, \beta$) are large compared to the inverse energy scale of interest (which is $m_p$), the eikonal exponentiation is expected to persist. This expectation is supported by the IR universality of the gravitational interaction, and it follows the logic that the eikonal resummation is dominated by ladder diagrams with soft graviton exchanges.

We acknowledge, however, that a full demonstration of exponentiation in quadratic gravity would require explicit computation of loop-level diagrams, particularly the sum of multi-graviton ladder diagrams. We intend to explore this in future work. This was first shown for Einstein gravity in the seminal work of 't Hooft [58] and then by others (see, e.g., [44]). Apart from gravitational field theories, for QFTs involving *massive* (relevant for our case) meson exchange the ladder structure can been nicely be identified and the S-matrix exponentiates [56]. There, the resummation of eikonal ladder diagrams leads to the well-known exponential form of the amplitude:

$$\mathcal{A}_{\text{eik}}(s, q_\perp) \sim 2s \int d^2 b \, e^{iq_\perp \cdot b} \left(e^{i\chi(s,b)} - 1\right), \tag{29}$$

where $\chi(s, b)$ is the eikonal phase and is given by

$$\chi(s, b) \sim \int d^D q \, \delta(2p_1 \cdot b) \delta(2p_2 \cdot b) \mathcal{A}_{\text{tree}}(s, -q^2), \quad \text{with } q^\mu = p_1^\mu - p_1^{\mu\prime}. \tag{30}$$

$b^\mu$ is the impact parameter. A recent review [57] discussed that this structure holds not only in General Relativity but also in UV-complete frameworks such as string theory. Given this, it is natural to expect that intermediate theories like quadratic gravity—lying between Einstein gravity and UV completions—should also admit eikonal exponentiation, at least in the IR regime. We refer specifically to Section 3.1.5 of [57], which supports this viewpoint. While a dedicated study of loop-level exponentiation in quadratic gravity is indeed warranted, we

believe the arguments above (as well as supported by our bottom-up computation presented in the manuscript) justify the assumption of exponentiation within the eikonal limit of the effective theory.

The eikonal phase for our case can be written in the following way,

$$
\begin{aligned}
\chi_{\text{EFT}} &= \frac{2\pi G}{Ep} \int \frac{d^2 k_\perp}{(2\pi)^2} e^{i\mathbf{k}_\perp \cdot \mathbf{x}_\perp} \left[ \widetilde{\mathbb{M}}_{\text{Born}}^{\text{EFT}}(s) + \frac{s^2}{k_\perp^2 + \mu^2 - i\epsilon} \right] \\
&= \underbrace{\frac{2\pi G \, \widetilde{\mathbb{M}}_{\text{Born}}^{\text{EFT}}(s)}{Ep} \int \frac{d^2 k_\perp}{(2\pi)^2} e^{i\mathbf{k}_\perp \cdot \mathbf{x}_\perp}}_{\text{Contact term contribution in quadratic EFT}} + \frac{2\pi G s^2}{Ep} \int \frac{d^2 k_\perp}{(2\pi)^2} e^{i\mathbf{k}_\perp \cdot \mathbf{x}_\perp} \frac{1}{k_\perp^2 + \mu^2 - i\epsilon} \\
&= \frac{2\pi G \, \widetilde{\mathbb{M}}_{\text{Born}}^{\text{EFT}}(s)}{Ep} \delta^{(2)}(\boldsymbol{x}_\perp) - \frac{Gs^2}{2Ep} \log(\mu|\boldsymbol{x}_\perp|).
\end{aligned}
\tag{31}
$$

Before proceeding with the detailed analysis of computing the eikonal amplitude, we would like to briefly pause and delve into an important conceptual detour. Specifically, we will discuss the notion of functions of the Dirac-$\delta$ function. This concept, although somewhat formal, plays a crucial role in interpreting expressions that arise in high-energy scattering, particularly in the eikonal approximation. A proper understanding of how to handle such expressions will be essential for the computations that follow.

---

**A digression on the functions of Dirac-$\delta$ function (measure)**

We now slightly deviate from our original discussion to briefly outline how one can meaningfully define functions of the $\delta$-function. As is well known, the $\delta$-function is a tempered distribution rather than a continuous probability distribution, as it has support only at a single point. Consequently, functions of such distributions are generally ill-defined. Nevertheless, as discussed in [59], it is possible—albeit with care—to give meaningful interpretations to functions of the $\delta$-function (especially the exponential map). We will briefly review this proposition, which, despite its subtleties, also provides a physically reasonable framework for our purposes.

The first approach proceeds as follows: one attempts to construct a resolvent-like function of the $\delta$-distribution to bypass difficulties associated with the exponential map. Consider the following object, interpreted as a distribution: $\frac{1}{1+\delta(x)}$.

**Proposition [59, 60].** *As a distribution,*

$$
\frac{1}{1 + \delta(x)} := \mathbb{I},
\tag{32}
$$

*where $\mathbb{I}$ denotes the Lebesgue measure. i.e, for each $\varphi(x) \in C_c^\infty(\mathbb{R})$, one has*

$$
\left\langle \frac{1}{1 + \delta(x)}, \varphi(x) \right\rangle := \int_{\mathbb{R}} \varphi(x).
\tag{33}
$$

While instructive, this proposition is of limited use in physical applications: it simply recovers the Lebesgue measure and entirely washes out the singular behaviour of the $\delta$-distribution. To address this, one must proceed more carefully. A more refined approach involves invoking the spectral theorem and exploiting the idempotence property of the $\delta$-function when treated as a characteristic set function [59].

> **Definition (Analytic Linearization)** [59]. *Let $f(\delta_a)$ be a function of the Dirac measure with singular support at $x = a$. Its analytic realization is given by the formal series expansion*
>
> $$f(\delta_a) \rightsquigarrow \mathbb{I} + \sum_{n=1}^{\infty} \frac{f^{(n)}(a)}{n!}\delta_a. \tag{34}$$
>
> This construction provides an analytic framework for defining functions of distributions. Importantly, it does not strictly contradict the earlier proposition. To see this, consider:
>
> $$\frac{1}{1+\delta_0} \rightsquigarrow \mathbb{I} - \delta_0(1-1+1-1+\dots) \overset{\text{reg}}{=} \mathbb{I} - \frac{1}{2}\delta_0, \tag{35}$$
>
> where the alternating sum $1-1+1-1+\dots$ – which, according to Riemann's reordering theorem, can be rearranged to converge to any real value—can be uniquely regularized via Dirichlet Eta summation yielding: $1-1+1-1+\dots \overset{\text{reg}}{=} \eta(0) = 1/2$. Thus, analytic linearization complements (or refines) the earlier proposition by preserving both the regular (Lebesgue) component and the singular structure encoded in Dirac-$\delta$, offering a physically meaningful extension well-suited to our purpose(s).

Now get back to our original discussion: the eikonal amplitude is given by

$$\mathcal{M}_{\text{eik}} = 8Ep \int d^2\boldsymbol{x}_\perp \, e^{-i\boldsymbol{q}\cdot\boldsymbol{x}_\perp} \left[ \exp\left(-\frac{iGs^2}{2Ep}\log(\mu|\boldsymbol{x}_\perp|)\right) \exp\left(\frac{2\pi i G\, \widetilde{\mathbb{M}}^{\text{EFT}}_{\text{Born}}(s)}{Ep}\delta^{(2)}(\boldsymbol{x}_\perp)\right) - 1 \right], \tag{36}$$

where, we have

$$\widetilde{\mathbb{M}}^{\text{EFT}}_{\text{Born}}(s) = -\frac{1}{48}s\left(\frac{(3\kappa + \alpha s - 8\beta s)}{(\kappa - \alpha s)(\kappa - 2\beta s)} - \frac{4}{\kappa + \alpha s} + \frac{1}{\kappa + 2\beta s}\right). \tag{37}$$

As seen from (36), the eikonal amplitude involves an integral over $e^{\delta(x)}$. Are they really bad or can we get some physically meaningful result out of it? Although functions of distributions are formally ill-defined, with careful treatment, $e^{\delta(x)}$ can be given a physically intuitive meaning. We use the 'Analytic Linearization' (34) to address this question. Now using this concept, we can rewrite (36) as[5]

$$\mathcal{M}_{\text{eik}} = 8Ep \int d^2\boldsymbol{x}_\perp \, e^{-i\boldsymbol{q}\cdot\boldsymbol{x}_\perp} \left[ (\mu|\boldsymbol{x}_\perp|)^{\frac{-iGs^2}{2Ep}} \left\{ 1 + (e-1)\frac{2\pi i G\widetilde{\mathbb{M}}^{\text{EFT}}_{\text{Born}}(s)}{Ep}\delta^{(2)}(\boldsymbol{x}_\perp) \right\} - 1 \right]. \tag{38}$$

We make use of the *Lebesgue measure and Schwartz bracket* [61] to extract the sensible piece. Furthermore, to evaluate the integral of (39) onwards we use the distributional nature of dirac delta function in terms of sharply picked gaussian distribution. Dividing the integral (38) into two parts we get

$$\begin{aligned}
\mathcal{M}^{\text{total}}_{\text{eik}} = {} & 8Ep \int d^2\boldsymbol{x}_\perp \, e^{-i\boldsymbol{q}\cdot\boldsymbol{x}_\perp} \exp\left(-\frac{iGs^2}{2Ep}\log(\mu|\boldsymbol{x}_\perp|)\right) \\
& + 8Ep \int d^2\boldsymbol{x}_\perp \, e^{-i\boldsymbol{q}\cdot\boldsymbol{x}_\perp}(e-1)\frac{2\pi i G\widetilde{\mathbb{M}}^{\text{EFT}}_{\text{Born}}(s)}{Ep}\exp\left(-\frac{iGs^2}{2Ep}\log(\mu|\boldsymbol{x}_\perp|)\right)\delta^{(2)}(\boldsymbol{x}_\perp) \\
:= {} & \mathcal{M}^{\text{GR}}_{\text{eik}} + \mathcal{M}^{\text{EFT}}_{\text{eik}},
\end{aligned} \tag{39}$$

---

[5]We use the following analytic linearization of exponential map to compute the celestial eikonal amplitude:

$$e^{\delta_0} \rightsquigarrow \mathbb{I} + \sum_{n=1}^{\infty} \frac{f^{(n)}(\delta_0 = 0)}{n!}\delta_0 = \mathbb{I} + (e-1)\delta_0.$$

where $\mathcal{M}_{\text{eik}}^{\text{GR}}$ and $\mathcal{M}_{\text{eik}}^{\text{EFT}}$ denote the GR part and the EFT correction respectively. In (39), the delta function simply evaluates $f(x_\perp)$ at $x_\perp = 0$, multiplying it by a constant, and has no effect at finite $x_\perp$. However, in the Regge limit, the dominant contribution to the amplitude arises from large transverse separations, $x_\perp$. Since the delta-function term is sharply localized at $x_\perp = 0$, it contributes only a constant to the eikonal amplitude. To obtain a physically meaningful result, we must smear the delta function around $x_\perp = 0$, as the logarithmic part of the amplitude is valid only in the regime of large $x_\perp$.

$$
\begin{aligned}
\mathcal{M}_{\text{eik}}^{\text{EFT}} &= 16\pi i G\, \widetilde{\mathbb{M}}_{\text{Born}}^{\text{EFT}}(s)(e-1)\lim_{\varepsilon \to 0} \int d^2 x_\perp\, e^{-i q \cdot x_\perp} (\mu |x_\perp|)^{\frac{-iGs^2}{2Ep}} \delta_\varepsilon^{(2)}(x_\perp) \\
&\to 16\pi i G\, \widetilde{\mathbb{M}}_{\text{Born}}^{\text{EFT}}(s)(e-1)\lim_{\varepsilon \to 0} \int dx_\perp d\theta\, x_\perp\, e^{-i q x_\perp \cos\theta} (\mu x_\perp)^{\frac{-iGs^2}{2Ep}} \frac{1}{x_\perp} \delta(\theta)\delta_\varepsilon(x_\perp) \quad (40) \\
&= \frac{16\pi i G\, \widetilde{\mathbb{M}}_{\text{Born}}^{\text{EFT}}(s)}{\sqrt{\pi}}(e-1)\Gamma\left(\frac{1}{2} - iGs\right)\left(\frac{1}{\mu\varepsilon}\right)^{2iGs}.
\end{aligned}
$$

From the above equation,[6] it is quite evident that after considering the distributional aspect of the contact delta function, we are able to modify the phase dressing in quadratic EFT along with an IR regulator $\mu_{IR} = \mu\varepsilon$. Now we proceed to show the analyticity properties, i.e. how the UV (or IR) behaviour of the eikonal amplitude has modified in the presence of the quadratic EFT corrections.

---

**Eikonal amplitude: Pure GR vs EFT correction**

$$
\mathcal{M}_{\text{eik}}^{\text{GR}} \sim -\underbrace{\frac{16\pi G \xi(s)}{t}}_{\text{Born Amplitude}} \times \underbrace{\frac{\Gamma(-iGs)}{\Gamma(iGs)}\left(\frac{4\mu^2}{-t}\right)^{-iGs}}_{\text{Phase}}
$$

$$
\oplus \tag{41}
$$

$$
\mathcal{M}_{\text{eik}}^{\text{EFT}} \sim \underbrace{\widetilde{\mathbb{M}}_{\text{Born}}^{\text{EFT}}(s|\alpha,\beta)}_{\text{Born amplitude}} \times \underbrace{\frac{16\pi G(e-1)}{\sqrt{\pi}}\frac{\Gamma(-2iGs)}{\Gamma(-iGs)}\left(\frac{2}{\mu\varepsilon}\right)^{2iGs}}_{\text{phase}}.
$$

As demonstrated in (41), EFT correction to the eikonal amplitude retains the same structural form as in GR. In particular, the eikonal amplitude continues to be expressed as the product of the Born amplitude and a phase factor. The principal distinction lies in the fact that, unlike in GR, the EFT-corrected eikonal amplitude is independent of the Mandelstam variable '$t$', thereby eliminating the associated '$z$'-dependence in the celestial eikonal amplitude which will be discussed in the next section.

---

## 3.3 Analyticity and dispersion relation for celestial eikonal amplitude

In this section, we explain the UV behaviour of the eikonal amplitude due to the presence of the higher curvature terms. We also inspect the IR behaviour for the quadratic EFT celestial amplitude. Below, we focus only on the contribution coming from the EFT (curvature

---

[6] The integral in Eq. (40) admits the following expansion in the smearing regulator $\varepsilon$:

$$
\mathcal{M}_{\text{eik}}^{\text{EFT}} \propto \left(\frac{1}{\mu\varepsilon}\right)^{2iGs}\left[\frac{1}{2}\Gamma\left(\frac{1}{2}-iGs\right) - \frac{1}{2}\varepsilon\, G |q_\perp| s\, \Gamma(-iGs) - \frac{1}{4}\varepsilon^2 |q_\perp|^2 \Gamma\left(\frac{3}{2}-iGs\right)\right].
$$

As evident, the amplitude depends on the Mandelstam variable $t = -q^2 = -|q_\perp|^2$ with the overall phase factor depending on the cutoff. However, this dependence appears only in higher orders of the $\varepsilon$ expansion and can thus be neglected for leading-order considerations.

squared) part. Although the celestial Born amplitude arising from EFT corrections is mero-morphic (as shown in (23))—rendering the construction of an eikonal amplitude technically unnecessary—employing eikonal exponentiation nonetheless offers a practical framework for capturing non-perturbative (in coupling) structures in the celestial amplitude, thereby address-ing the second question posed in the introduction. However, the eikonal amplitude encodes the full non-perturbative scattering amplitude within the given eikonal regime.

*UV behaviour* : Thus, the $2 \to 2$ scattering amplitude of massless scalars as external states can be casted with the contribution of phase in the eikonal limit as follows,[7]

$$\mathcal{A}_{eik}^{\text{EFT}}(\gamma, z) = \frac{16\pi i G}{\sqrt{\pi}}(e-1)2^{3-\gamma}z^2 \int_0^\infty d\omega\,\omega^{\gamma-1}\,\Gamma\left(\frac{1}{2}-iG\omega^2\right)\left(\frac{1}{\mu\varepsilon}\right)^{2iG\omega^2} \widetilde{\mathbb{M}}_{\text{Born}}^{\text{EFT}}(\omega^2)$$
$$+ \text{GR part.} \tag{42}$$

In the large $\omega$ limit, the integral reduces to

$$\mathcal{A}_{eik}^{\text{EFT}}(\gamma, z) = \frac{i\pi G g(\alpha,\beta)}{\sqrt{\pi}}(e-1)\,2^{7-\gamma}z^2 \int_0^\infty d\omega\,\omega^{\gamma-1}\,\Gamma\left(\frac{1}{2}-iG\omega^2\right)\left(\frac{1}{\mu\varepsilon}\right)^{2iG\omega^2}$$
$$+ \text{GR part} \tag{43}$$
$$= i\pi G g(\alpha,\beta)(e-1)2^{8-\gamma}z^2 \int_0^\infty d\omega\,\omega^{\gamma-1}\left(\frac{2}{\mu\varepsilon}\right)^{2iG\omega^2}\frac{\Gamma\left(-2iG\omega^2\right)}{\Gamma\left(-iG\omega^2\right)} + \dots,$$

where $g(\alpha,\beta) = \frac{8\beta-\alpha}{2\alpha\beta} + \frac{4}{\alpha} - \frac{1}{2\beta}$. In obtaining the second line, we used the duplication formula, $\frac{\Gamma(a)\Gamma(a+\frac{1}{2})}{\Gamma(2a)} = \frac{\sqrt{\pi}}{2^{2a-1}}$. Now we have

$$\mathcal{A}_{eik}^{\text{EFT}}(\gamma, z) = i\pi G g(\alpha,\beta)(e-1)2^{7-\gamma}z^2 \int_0^\infty dx\,x^{\frac{\gamma-2}{2}}\left(\frac{2}{\mu\varepsilon}\right)^{2iGx}\frac{\Gamma(-2iGx)}{\Gamma(-iGx)}$$
$$\xrightarrow{x\to\infty} i\pi G g(\alpha,\beta)(e-1)2^{7-\gamma}z^2\sqrt{2}\int_0^\infty dx\,x^{\frac{\gamma-2}{2}}\left(\frac{e}{\mu^2\varepsilon^2}\right)^{iGx}(-iGx)^{-iGx}. \tag{44}$$

The integral[8] can be done by analytically continuing it in the fourth quadrant of the complex plane.

$$\mathcal{A}_{eik}^{\text{EFT}}(\gamma, z) = i\pi G g(\alpha,\beta)(e-1)2^{\frac{15}{2}-\gamma}z^2\left(-\frac{i}{G}\right)^{\gamma/2-1}\int_0^\infty d\zeta\,\zeta^{\frac{\gamma}{2}-1}\left(\frac{e}{\mu^2\varepsilon^2}\right)^{\zeta}(-\zeta)^{-\zeta}. \tag{45}$$

The integral in (45) is analytic for carefully chosen IR cut-off, more precisely the integral is meromorphic in the complex $\gamma$ plane. For large $\gamma$, the integral in (45) can be done using the saddle point approximation[9]

$$\mathcal{A}_{eik}^{\text{EFT}}(\gamma, z) \xrightarrow{\gamma\gg1} i\pi G g(\alpha,\beta)(e-1)2^{\frac{15}{2}-\gamma}z^2\left(-\frac{i}{G}\right)^{\gamma/2}e^{f(\zeta_*)}\sqrt{\frac{\pi\zeta_*^2}{2\zeta_*+\gamma}}, \tag{46}$$

where, $\zeta_* = \frac{\gamma}{2W\left(-\frac{1}{2}e\gamma\mu^2\varepsilon^2\right)}$ and $f(\zeta) = \frac{1}{2}\gamma\log(\zeta) - \zeta\log(-\zeta) - \zeta\log\left(\mu^2\varepsilon^2\right) + 1$, with $W$ being the Lambert-$W$ function.

---

[7]In the argument of $\Gamma$ function, we neglect terms like $G\alpha \sim \ell_p^2 \ll 1$.

[8]We found a mismatch between signs of $a$ and $b$ in the expression $e^{ax}x^{ibx}$ in [41] where it seems to be of opposite sign according to the asymptotic behaviour of the ratio of gamma functions.

[9]Though it is tough to compare the convergence rate of the integral to that of its GR counterpart, as it is done under the saddle-point approximation, we have checked both integrals numerically and found that the UV behaviour of the part coming from the quadratic gravity is comparatively better.

**A sidebar**   A quick question that may arise is how one can justify expanding the integrand in (42) in the large-$\omega$ limit, given that the integration is performed over the entire range $(0, \infty)$. The explanation goes as follows: The expansion of the integrand in the large-$\omega$ limit is employed to extract the ultrablack (UV) behavior of the corresponding celestial amplitude. Strictly speaking, the appropriate procedure is to first expand the integrand in the large-$\omega$ regime, perform the indefinite integral, and then analyze the $\omega \to \infty$ behavior of the result.

Nonetheless, guided by intuition from the saddle point approximation, one expects that in the presence of a large parameter, the dominant contribution to the integral originates near the peak of the integrand, where it is sharply localized. Although the integration domain extends over the full range $(0, \infty)$, the leading contribution arises from the large-$\omega$ region, which justifies the use of this approximation in capturing the UV behavior. This is why we did not evaluate the integral in (42) explicitly; instead, we applied a saddle point analysis by choosing the parameter $\gamma$ to be large, leading to a dominant contribution near the saddle point $\zeta_*$.

For a more rigorous justification, one could invoke the "strategy of regions" a well-established technique widely used in the computation of multi-loop Feynman integrals [62]. This method systematically *expands the integrand* in distinct regions characterized by separated scales, allowing for controlled approximations. In our context, this methodology has been applied heuristically, in line with earlier treatments in [41].

**IR behaviour**   : Now, to investigate the analytic property of the eikonal amplitude in IR limit, we first consider a general integral of the form

$$\mathcal{I}(a) := \int_0^\infty dx\, x^{a-1}\, \phi(x) c^{\ell x}, \qquad \mathrm{Re}(\ell) = 0\,, \tag{47}$$

where $a \in \mathbb{C}$ and c is a constant. Now $\phi(x)$ is analytic around $x = 0$. Also, it doesn't affect the convergence at infinity. Now, to examine the IR behaviour, we expand $c^{\ell x}$ and $\phi$ in Taylor series around $x = 0$ and see what happens to the integral.[10] The integral becomes

$$\begin{aligned}
\mathcal{I}_L(a) &= \sum_{m,n=0}^\infty \phi_m \frac{\ell^n \log(c)^n}{n!} \int_0^L dx\, x^{a+m+n-1} \\
&= \sum_{m,n=0}^\infty \phi_m \frac{\ell^n \log(c)^n}{n!} \frac{L^{a+m+n}}{(a+m+n)} \sim \sum_{m,n=0}^\infty \frac{\ell^n \log(c)^n \phi_m}{n!(a+n+m)} L^{a+m+n} + \text{regular terms.}
\end{aligned} \tag{48}$$

The integral has poles and admits the following expansion (with $k = m + n$),

$$\mathcal{I}_k(a) \sim \frac{\ell^k \log(c)^k L^{a+k} \phi_0}{k!(a+k)} + \frac{\ell^{k-1} \log(c)^{k-1} L^{a+k} \phi_1}{(k-1)!(a+k)} + \dots \text{regular.} \tag{49}$$

Now, in the actual eikonal amplitude given in (43), has Gamma functions which can be expanded in a Taylor series (near $\omega = 0$) as follows

$$\log \Gamma\left(\frac{1}{2} - iG\omega\right) = \gamma_E\left(\frac{1}{2} + iG\omega\right) + \sum_{p \geq 2} \frac{\zeta(p)}{p}\left(\frac{1}{2} + iG\omega\right)^p, \tag{50}$$

where $\zeta(p)$ is the Riemann zeta function and $\gamma_E$ is the Euler's constant. Therefore, the eikonal amplitude for our case can be written as

$$\mathcal{A}_{eik}^{\mathrm{EFT}}(\gamma, z) \sim C_E \int_0^\infty d\omega'\, \omega'^{\gamma/2-1} e^{ib\omega'} \exp\left[\sum_{p \geq 2} \frac{\zeta(p)}{p}\left(\frac{1}{2} + iG\omega'\right)^p\right] \widetilde{\mathbb{M}}_{\mathrm{Born}}^{\mathrm{EFT}}(\omega', z), \tag{51}$$

---

[10]Here, $L > 0$ but a small number.

where

$$b = G\left[\log\left(\frac{4}{\mu^2\varepsilon^2}\right) + \gamma_E\right], \qquad C_E = e^{\gamma_E/2}.$$ (52)

Now by comparing term by term with (49), we can readily identify

$$
\begin{aligned}
\phi_0 &= 0, \\
\phi_1 &= \frac{1}{12}\sqrt{\pi}G^2\omega(2\alpha - \beta), \\
\phi_2 &= -\frac{1}{12}i\sqrt{\pi}G^3(\gamma_E + \log(4))(2\alpha - \beta), \\
\phi_3 &= -\frac{1}{48}\sqrt{\pi}G^4\left(-8\alpha^3 + (\gamma_E + \log(4))^2(4\alpha - 2\beta) + 2\pi^2\alpha + 16\beta^3 - \pi^2\beta\right), \\
\phi_4 &= -\frac{1}{144}i\sqrt{\pi}G^5\left(24\psi^{(0)}\left(\frac{1}{2}\right)(\alpha^3 - 2\beta^3)\right. \\
&\qquad\qquad \left. -\left(3\pi^2\psi^{(0)}\left(\frac{1}{2}\right) + 2\left(\psi^{(0)}\left(\frac{1}{2}\right)^3 + \psi^{(2)}\left(\frac{1}{2}\right)\right)\right)(2\alpha - \beta)\right),
\end{aligned}
$$ (53)

$$\vdots$$

where $\psi^{(0)}(x)$ is the digamma function and $\psi^{(2)}(x)$ is the second-order derivative of the digamma function. Clubbing together all the pieces, we get

$$\mathcal{A}_{eik}^{\text{EFT}}(\gamma, z) \sim \left(\frac{L^{\frac{\gamma}{2}+n+1}\left(2iG\log(\frac{2}{\mu\varepsilon})\right)^n \phi_1}{n!(\frac{\gamma}{2}+n+1)} + \ldots + \text{regular contribution}\right).$$ (54)

It is evident that, in the IR limit, the leading singularity coming from the EFT contribution differs from that of pure GR. As evident from (54), the location of the leading pole is at $\gamma = -2(n+1)$, and it is now a simple pole, unlike GR, where we get contributions from higher order poles. The introduction of higher curvature terms changes the infrared behaviour by changing only the subleading pole (i.e, simple pole) structure. Now, we proceed to discuss the dispersion relation for the quadratic EFT.

## 3.4 Dispersion relations

Although, like GR [41] we are unable to compute the integral in eikonal amplitude explicitly, we still find out the dispersion relation. The dispersion relation can be figured out from the analytic continuation of the integral (43) in the full complex plane. To do so, we define the eikonal amplitude with an appropriate $i\varepsilon$-prescription,[11]

$$\mathcal{A}_{eik}^{\text{EFT}}(\gamma, z) = \frac{16\pi iG}{\sqrt{\pi}}(e-1)2^{3-\gamma}z^2 \int_{0-i\epsilon}^{\infty-i\epsilon} d\omega\, \omega^{\gamma-1}\Gamma\left(\frac{1}{2} - iG\omega^2\right)\left(\frac{1}{\mu\varepsilon}\right)^{2iG\omega^2}\widetilde{\mathbb{M}}_{\text{Born}}^{\text{EFT}}(\omega),$$ (55)

where, $\widetilde{\mathbb{M}}_{\text{Born}}^{\text{EFT}}$ is defined in (37).

The integral can also be written after a change of variable change in the following way,

$$\sim z^2 \int_{0-i\epsilon}^{\infty-i\epsilon} d\omega'\, \omega'^{\gamma/2}\Gamma\left(\frac{1}{2} - iG\omega'\right)\left(\frac{1}{\mu\varepsilon}\right)^{2iG\omega'}\frac{\widetilde{\mathbb{M}}_{\text{Born}}^{\text{EFT}}(\omega')}{\omega'}.$$ (56)

---

[11]A natural question arises regarding the choice of the $-i\epsilon$-prescription. This choice is essential to impose the correct limiting conditions to recover the results for GR. In contrast, adopting a $+i\epsilon$ prescription causes the integration contour to encircle certain unphysical poles whose residues become divergent in the limit $\alpha, \beta \to 0$. To ensure consistency and recover GR result in the limit $\alpha, \beta \to 0$, it is therefore necessary to employ the $-i\varepsilon$ prescription.



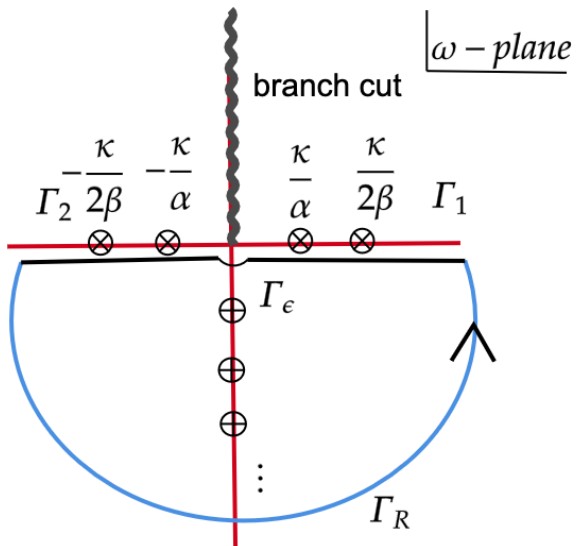

Figure 4: Figure depicting the chosen contour for dispersion relation in the complex-$\omega$ plane. The cross signs depict the location of the poles.

This has poles at

$$G\omega' = e^{-i\pi/2}\left(n + \frac{1}{2}\right), \quad \text{and} \quad \omega' = \frac{\kappa}{\alpha}, -\frac{\kappa}{\alpha}, \frac{\kappa}{2\beta}, -\frac{\kappa}{2\beta}, \quad n \in \mathbb{Z}_{\geq 0}. \tag{57}$$

Note that we also have extra poles at $\omega'$ entirely due to the higher curvature contributions. We also have a branch cut, which is running from $0 \to i\infty$. However, as we have poles on positive and negative real axis, we need to change the argument about the branch cut. The integral in $\omega'$ can now be analytically continued to the complex plane, and we use a contour in the clockwise orientation in the complex plane as depicted in Fig. 4. The integral can now be written as a sum of four separate pieces as

$$I_C \sim z^2 \oint_C d\omega\, \omega^{\gamma/2} \Gamma\left(\frac{1}{2} - iG\omega\right)\left(\frac{1}{\mu\varepsilon}\right)^{2iG\omega} \frac{\widetilde{\mathbb{M}}_{\text{Born}}^{\text{EFT}}(\omega)}{\omega}$$
$$= I_1 + I_2 + I_\epsilon + I_R, \tag{58}$$

where the decomposition of the contour $\Gamma = \Gamma_1 \cup \Gamma_R \cup \Gamma_2 \cup \Gamma_\epsilon$ corresponds to each of the four integrals in the second line, with $\Gamma_R$ the semi-circle part of the contour of radius $R$ and $\Gamma_\epsilon$ the smaller semi-circle part of radius $\epsilon$ as shown in Fig. (4). It can be shown that in the limits where $\epsilon \to 0$ and $R \to \infty$, the contributions from $\Gamma_\epsilon$ and $\Gamma_R$ vanishes. Therefore, we have

$$I_1 + I_2 = 2\pi i \sum_{n \geq 1} \text{Res}\left[z^2\, \omega^{\gamma/2} \Gamma\left(\frac{1}{2} - iG\omega\right)\left(\frac{1}{\mu\varepsilon}\right)^{2iG\omega} \frac{\widetilde{\mathbb{M}}_{\text{Born}}^{\text{EFT}}(\omega)}{\omega}\right], \tag{59}$$

at $\omega = G^{-1}e^{-i\pi/2}(n + 1/2)$, where the integrals $(I_1, I_2)$ are given by

$$I_1 = -\mathcal{A}_{eik}^{\text{EFT}}(\gamma, z), \qquad I_2 = e^{-i\pi\gamma/2}\, \overline{\mathcal{A}_{eik}^{\text{EFT}}}(\bar{\gamma}, -z). \tag{60}$$

$\overline{\mathcal{A}_{eik}^{\text{EFT}}}(\bar{\gamma})$ denotes the complex conjugate of the eikonal amplitude $\mathcal{A}_{eik}^{\text{EFT}}(\gamma)$. Therefore we can

write the following *dispersion relation* including the GR contribution as

$$-\left(\mathcal{A}_{eik}^{\mathrm{GR}}(\gamma,z)+\mathcal{A}_{eik}^{\mathrm{EFT}}(\gamma,z)\right)+e^{-i\pi\gamma/2}\left(\overline{\mathcal{A}_{eik}^{\mathrm{GR}}}(\bar{\gamma},-z)+\overline{\mathcal{A}_{eik}^{\mathrm{EFT}}}(\bar{\gamma},-z)\right)$$

$$=z^2\sum_{n\geq 1}\frac{(-1)^n\frac{16\pi iG}{\sqrt{\pi}}(e-1)2^{3-\gamma}2^{-4}\left(-\frac{i\left(n+\frac{1}{2}\right)}{G}\right)^{\gamma/2}}{3n!(-\alpha-2i-2\alpha n)(-\alpha+2i-2\alpha n)(-\beta-i-2\beta n)(-\beta+i-2\beta n)}$$

$$\times\left[(2n+1)\mu_{\mathrm{IR}}^{-2n-1}\left(4(-\beta+2\alpha)+\alpha\beta(2n+1)^2(-\alpha+8\beta)\right)\right] \tag{61}$$

$$-\frac{\pi z^3 2^{3-\gamma}}{z-1}\sum_{k\geq 1}\frac{i^k}{k!(k-1)!}\left(-\frac{ik}{G}\right)^{\gamma/2}\left(\frac{k(z-1)}{4zG\mu^2}\right)^k.$$

The sum in (61) does not admit a closed-form expression and is therefore left as a formal sum, where, $\mu_{IR}=\mu\varepsilon$ is a dimensionless IR regulator. As a consistency check, one can verify that setting $\alpha=\beta=0$ recovers the result corresponding to GR [41]. Up to this point, we have investigated the analyticity and pole structure of the celestial amplitude, highlighting the modifications due to the incorporation of higher curvature terms. In the following section, we focus on the properties of the conformal four-point function that follows from it.

Subsequently, we compute the shadow transform of the celestial four-point correlator of primary operators arising from the quadratic effective field theory (EFT). We then carry out the conformal block decomposition and determine the associated operator product expansion (OPE) coefficients.

# 4 Shadowed correlator and operator product expansion (OPE)

As introduced in [3,13] and briefly discussed in the introduction, there is significant progress in understanding the 2D holographic description of 4D scattering amplitude in flat space. Specifically, the action of Lorentz group $SL(2,\mathbb{C})$ on the kinematic data can be rescasted as the Mobius transformation on the celestial sphere and the scattering amplitudes (denoted as $\mathcal{A}_n(z_i,\bar{z}_i)$), defined on the boost eigenstate, can be re-interpreted as correlation function in a 2D CFT.

In light of this, in this section, our aim is to explore the structure of the celestial conformal field theory (CCFT) associated with the quadratic effective field theory (EFT) amplitude (in the eikonal limit) in flat space. A crucial step in this direction is to determine the full operator spectrum of the conformal theory on the celestial sphere. This, in turn, requires knowledge of the OPE coefficients in various channels. Primarily, we focus on the shadowed amplitude. We first employ the Burchnall-Chaundy (BC) expansion to extract the OPE coefficients by comparing it with the conformal block expansion of Dolan and Osborn [63]. However, we find that the BC expansion appears insufficient to capture the complete spectrum of the theory. To go beyond this limitation, we subsequently incorporate the effects of spinning exchanges (which completes the spectrum) in the OPE coefficients using the Euclidean OPE inversion formula [29–31,64] and consequently comment on the subtlety associated with the inversion.

The *shadow* of a conformal primary with conformal dimension $h,\bar{h}$ (and scaling dimension $\Delta=h+\bar{h}$) can be written as [28,65]

$$\widetilde{\mathcal{O}}_{\tilde{\Delta}}(z,\bar{z}):=\widetilde{\mathcal{O}_{\Delta}(y,\bar{y})}=\mathcal{K}_{h,\bar{h}}\int d^2y(z-y)^{2h-2}(\bar{z}-\bar{y})^{2\bar{h}-2}\mathcal{O}_{\Delta}(y,\bar{y}), \tag{62}$$

where the constant is given by

$$\mathcal{K}_{h,\bar{h}}=\frac{(-1)^{2(h-\bar{h})}\Gamma(2-2h)}{\pi\Gamma(2\bar{h}-1)}. \tag{63}$$

The shadow operator is a primary with conformal dimensions $\{1-h, 1-\bar{h}\}$ which corresponds to a scaling dimension $\tilde{\Delta} = 2 - \Delta$.

## 4.1 Shadowed amplitude corresponding to eikonal amplitude

The 4-point eikonal amplitude is given by

$$\mathcal{M}_{\text{eik}}(z_i, \bar{z}_i) = \lim_{z_4 \to \infty} \frac{1}{|z|^{\Delta_1 + \Delta_2} |z_4|^{2\Delta_4}} (z-1)^{\frac{\Delta_1 - \Delta_2 - \Delta_3 + \Delta_4}{2}} \delta(iz - i\bar{z}) \mathcal{M}_{\text{eik}}(\boldsymbol{\Delta}, z). \tag{64}$$

The normalized amplitude is given by

$$\widehat{\mathcal{M}}_{\text{eik}} = \lim_{z_4, \bar{z}_4 \to \infty} z_4^{\Delta_4} \bar{z}_4^{\Delta_4} \mathcal{A}_{\text{eik}}(z_i, \bar{z}_i). \tag{65}$$

Therefore the shadowed amplitude is given by

$$\widetilde{\mathcal{M}}_{\text{eik}}^{\text{EFT}}(w, \bar{w}) = \mathcal{K}_{h_2, \bar{h}_2} \lim_{z_4 \to \infty} |z_4|^{2\Delta_4} \int \frac{d^2 z}{(z-w)^{\Delta_2} (\bar{z} - \bar{w})^{2-\Delta_2}} \frac{1}{|z|^{\Delta_1 + \Delta_2}} (z-1)^{\frac{\Delta_1 - \Delta_2 - \Delta_3 + \Delta_4}{2}} \delta(iz - i\bar{z})$$

$$\times i\pi G \widetilde{\mathbb{M}}_{\text{Born}}^{\text{EFT}} (e-1) 2^{8-\gamma} z^2 \int_0^\infty d\omega\, \omega^{\gamma-1} \left(\frac{2}{\mu\varepsilon}\right)^{2iG\omega^2} \frac{\Gamma(-2iG\omega^2)}{\Gamma(-iG\omega^2)}. \tag{66}$$

Here $\widetilde{M}_{\text{Born}}^{\text{EFT}}(\omega)$ is defined in (37). It contains only the terms arising due the presence of the curvature squared terms. Note that in equation (66), the $\omega$-integral is independent of the cross-ratio $z$. This allows us to perform the integrals over $z$ and $\bar{z}$ explicitly, resulting in a function that depends only on $w$, $\bar{w}$, and the remaining integral over $\omega$. Consequently, the overall structure of the shadow amplitude (coming from the curvature squared terms) remains unchanged if we replace the full non-perturbative (in coupling) eikonal amplitude with just the Born amplitude. Hence, from the following section onward, we focus on the Born amplitude to study the conformal correlator and its operator product expansion.

Before we concluding we note that, for the GR part, the computation of the shadow amplitude is not so straightforward. The eikonal shadow amplitude for GR has the following form:

$$\widetilde{\mathcal{M}}_{\text{eik}}^{\text{GR}}(w, \bar{w}) = \mathcal{K}_{h_2, \bar{h}_2} \lim_{z_4 \to \infty} |z_4|^{2\Delta_4} \int \frac{d^2 z}{(z-w)^{\Delta_2} (\bar{z} - \bar{w})^{2-\Delta_2}} \frac{1}{|z|^{\Delta_1 + \Delta_2}} (z-1)^{\frac{\Delta_1 - \Delta_2 - \Delta_3 + \Delta_4}{2}} \delta(iz - i\bar{z})$$

$$\times \left(\frac{Gz}{1-z}\right) \int_0^\infty d\omega\, \omega^{\gamma+1} \left(\frac{4z\mu^2}{\omega^2(z-1)}\right)^{-iG\omega^2} \frac{\Gamma(-iG\omega^2)}{\Gamma(iG\omega^2)}. \tag{67}$$

As can be seen easily from (67), the integral over $z$ and $\bar{z}$ can be performed explicitly, leaving us with an integral over $\omega$. However, this remaining integral cannot be evaluated analytically, making it difficult to obtain the shadowed amplitude non-perturbatively in $G$ for GR. Our primary objective is to extract the OPE coefficient analytically from the conformal block expansion, which requires a closed-form expression (at least in $w$, $\bar{w}$) for the amplitude. Fortunately, the contribution due to the EFT correction in the Born amplitude possesses a well-behaved analytic structure, allowing us to bypass the need for eikonal resummation in both qualitative and quantitative analyses. Therefore, in the following subsections, we will consider only the Born amplitude when computing the OPE coefficients.

## 4.2 Shadowed amplitude corresponding to celestial Born amplitude

We start by reminding that the 4-point amplitude is given by

$$\mathcal{A}_4(z_i, \bar{z}_i) = \lim_{z_4 \to \infty} \frac{1}{|z|^{\Delta_1 + \Delta_2} |z_4|^{2\Delta_4}} (z-1)^{\frac{\Delta_1 - \Delta_2 - \Delta_3 + \Delta_4}{2}} \delta(iz - i\bar{z}) \mathcal{A}(\boldsymbol{\Delta}, z). \tag{68}$$

Consequently, one can define the normalized 4-point amplitude as

$$\widehat{\mathcal{A}_4}(z_i, \bar{z}_i) = \lim_{z_4, \bar{z}_4 \to \infty} z_4^{\Delta_4} \bar{z}_4^{\Delta_4} \mathcal{A}_4(z_i, \bar{z}_i). \tag{69}$$

Our goal is to compute the celestial amplitude of the shadowed correlator, find the conformal block expansion and correspondingly calculate the partial wave coefficients using *Euclidean OPE inversion formula*. Now, we want to compute the following correlator,

$$\widetilde{\mathcal{A}_4}(w, \bar{w}) := \lim_{z_4 \to \infty} |z_4|^{2\Delta_4} \left\langle \mathcal{O}_{\Delta_1}(0,0) \widetilde{\mathcal{O}}_{2-\Delta_2}(w, \bar{w}) \mathcal{O}_{\Delta_3}(1,1) \mathcal{O}_{\Delta_4}(z_4, \bar{z}_4) \right\rangle \tag{70}$$

$$= \mathcal{K}_{h_2, \bar{h}_2} \lim_{z_4 \to \infty} |z_4|^{2\Delta_4} \int \frac{d^2 z}{(z-w)^{\Delta_2}(\bar{z}-\bar{w})^{2-\Delta_2}} \left\langle \mathcal{O}_{\Delta_1}(0,0) \mathcal{O}_{\Delta_2}(z, \bar{z}) \mathcal{O}_{\Delta_3}(1,1) \mathcal{O}_{\Delta_4}(\bar{z}_4, \bar{z}_4) \right\rangle.$$

We can define three celestial amplitudes in the respective kinematic region as we have the integral over the cross-ratio $z$ as given in Table (1).

$$\mathcal{K}_{h_2, \bar{h}_2}^{-1} \widetilde{\mathcal{A}_4}(w, \bar{w}) = \int \frac{dz}{(z-w)^{2-\Delta_2}(z-\bar{w})^{2-\Delta_2}} \frac{(z-1)^{\frac{\Delta_1-\Delta_2-\Delta_3+\Delta_4}{2}}}{|z|^{\Delta_1+\Delta_2}} \mathcal{A}(\boldsymbol{\Delta}, z). \tag{71}$$

The integrals can be expressed in terms of *Appell hypergeometric* functions and the amplitude for the different kinematic channels take the following form:[12]

***12→34 kinematics:*** By defining $w = \frac{z_{12'} z_{34}}{z_{13} z_{2'4}}$ we can cast the integral in (71) for the $12 \to 34$ kinematics in the following way,[13]

$$\mathcal{K}_{h_2, \bar{h}_2}^{-1} \widetilde{\mathcal{A}_4}^{12 \to 34}(w, \bar{w})$$

$$= \int_1^\infty \frac{dz}{(z-w)^{2-\Delta_2}(z-\bar{w})^{2-\Delta_2}} \frac{(z-1)^{\frac{\Delta_1-\Delta_2-\Delta_3+\Delta_4}{2}}}{|z|^{\Delta_1+\Delta_2}} \mathcal{A}^{12 \to 34}(\boldsymbol{\Delta}, z)$$

$$= \int_1^\infty \frac{dz}{(z-w)^{2-\Delta_2}(z-\bar{w})^{2-\Delta_2}} \frac{(z-1)^{\frac{\Delta_1-\Delta_2-\Delta_3+\Delta_4}{2}}}{|z|^{\Delta_1+\Delta_2}} \left( \delta_1(\alpha, \beta|\gamma) z^2 + \delta_2(\gamma) \frac{z^3}{z-1} \right) \tag{72}$$

$$= \delta_1(\alpha, \beta|\gamma) B\left( \frac{1}{2}\left(2+\Delta_1-\Delta_2+\Delta_3-\Delta_4\right), \frac{1}{2}\left(2+\Delta_1-\Delta_2-\Delta_3+\Delta_4\right) \right)$$

$$\times F_1\left( \frac{1}{2}\left(2+\Delta_1-\Delta_2+\Delta_3-\Delta_4\right), 2-\Delta_2, 2-\Delta_2, 2+\Delta_1-\Delta_2, w, \bar{w} \right)$$

$$+ \text{(contribution from GR)}.$$

---

[12]
$$\mathbf{F}_1(a, b_1, b_2, c, x, y) = \sum_{m,n=0}^{\infty} \frac{(a)_{m+n}(b_1)_m(b_2)_n}{(c)_{m+n} m! n!} x^m y^n, \qquad \max(|x|, |y|) < 1,$$

where $(a)_k$ is the Pochhammer symbol is given by, $(a)_k := \frac{\Gamma(a+k)}{\Gamma(a)}$. Also the integral representation of the same function is given by

$$\mathbf{F}_1(a, b_1, b_2, c, x, y) = \frac{1}{B(a, c-a)} \int_0^1 dm \frac{m^{a-1}(1-m)^{c-a-1}}{(1-mx)^{b_1}(1-my)^{b_2}}, \qquad \mathcal{R}(c) > 0, \quad \text{and} \quad \mathcal{R}(c-a) > 0,$$

with, $B(a, b) := \frac{\Gamma(a)\Gamma(b)}{\Gamma(a+b)}$, $B(a, b) = \int_0^1 dx\, x^{a-1}(1-x)^{b-1}$ is the usual Euler Beta function. Given these, we now proceed to compute the shadowed four-point correlator in three different kinematic regimes.

[13]For four general points $z_1, z_2', z_3, z_4$. $z_2'$ is the location of the insertion of the shadow primary.

The contribution from GR can be straightforwardly computed as

(contribution from GR)

$$= \delta_2(\gamma) \int_1^\infty \frac{dz}{(z-w)^{2-\Delta_2}(z-\bar{w})^{2-\Delta_2}} \frac{(z-1)^{\frac{\Delta_1-\Delta_2-\Delta_3+\Delta_4}{2}-1}}{|z|^{\Delta_1+\Delta_2-3}}$$

$$\xrightarrow{z\to 1/z} \delta_2(\gamma) \int_0^1 \frac{dz}{(1-zw)^{2-\Delta_2}(1-z\bar{w})^{2-\Delta_2}} (1-z)^{\frac{\Delta_1-\Delta_2-\Delta_3+\Delta_4}{2}-1} z^{\frac{1}{2}(\Delta_1-\Delta_2+\Delta_3-\Delta_4)} \quad (73)$$

$$= \delta_2(\gamma) B\left(\frac{1}{2}(\Delta_1-\Delta_2+\Delta_3-\Delta_4+2), \frac{1}{2}(\Delta_1-\Delta_2-\Delta_3+\Delta_4)\right)$$

$$\times F_1\left(\frac{1}{2}(\Delta_1-\Delta_2+\Delta_3-\Delta_4+2), 2-\Delta_2, 2-\Delta_2, \Delta_1-\Delta_2+1, w, \bar{w}\right).$$

In the later computations we will not explicitly show the contributions from GR as our primary goal to examine the contribution coming from the quadratic corrections. Hence, finally the 4-point correlator for the single shadowed operator(s) in the $\mathfrak{s}$-channel can be written as

$$\lim_{z_4\to\infty} |z_4|^{2\Delta_4} \left\langle \mathcal{O}_{\Delta_1}(0,0)\widetilde{\mathcal{O}_{\Delta_2}}(w,\bar{w})\mathcal{O}_{\Delta_3}(1,1)\mathcal{O}_{\Delta_4}(z_4,\bar{z}_4)\right\rangle = \frac{1}{|w|^{2+\Delta_1-\Delta_2}}\mathcal{G}^{\mathfrak{s}}(w,\bar{w}). \quad (74)$$

Therefore we can identify the conformal block corresponding to quadratic EFT to be

$$\mathcal{G}^{12\to34}(w,\bar{w}) = \delta_1(\alpha,\beta|\gamma)|w|^{2+\Delta_1-\Delta_2} B\left(\frac{1}{2}(2+\Delta_1-\Delta_2+\Delta_3-\Delta_4), \frac{1}{2}(2+\Delta_1-\Delta_2-\Delta_3+\Delta_4)\right)$$

$$\times F_1\left(\frac{1}{2}(2+\Delta_1-\Delta_2+\Delta_3-\Delta_4), 2-\Delta_2, 2-\Delta_2, 2+\Delta_1-\Delta_2, w, \bar{w}\right) \quad (75)$$

$$+ (\text{contribution from GR}).$$

***13→24 kinematics:*** The $\mathfrak{t}$-channel contribution to the four-point function in quadratic EFT is given by[14]

$$\mathcal{G}^{13\to24}(w,\bar{w}) = \int_0^1 \frac{dz}{(z-w)^{2-\Delta_2}(z-\bar{w})^{2-\Delta_2}} \frac{(z-1)^{\frac{\Delta_1-\Delta_2-\Delta_3+\Delta_4}{2}}}{|z|^{\Delta_1+\Delta_2}} \mathcal{A}^{13\to24}(\mathbf{\Delta}, z)$$

$$= \delta_1(\alpha,\beta|\gamma)(w\bar{w})^{2-\Delta_2} B\left(-\Delta_1-\Delta_2+3, \frac{1}{2}(\Delta_1-\Delta_2-\Delta_3+\Delta_4+2)\right) \quad (76)$$

$$\times F_1\left(3-\Delta_1-\Delta_2, 2-\Delta_2, 2-\Delta_2, \frac{1}{2}(-\Delta_1-3\Delta_2-\Delta_3+\Delta_4+8), \frac{1}{w}, \frac{1}{\bar{w}}\right).$$

***14→23 kinematics:*** Similarly for the $14 \to 23$ kinematics ($\mathfrak{u}$-channel), four-point function has the following form,

$$\mathcal{G}^{14\to23}(w,\bar{w}) = \int_{-\infty}^0 \frac{dz}{(z-w)^{2-\Delta_2}(z-\bar{w})^{2-\Delta_2}} \frac{(z-1)^{\frac{\Delta_1-\Delta_2-\Delta_3+\Delta_4}{2}}}{|z|^{\Delta_1+\Delta_2}} \mathcal{A}^{14\to23}(\mathbf{\Delta}, z). \quad (77)$$

Now changing the integral variable as $z \to \frac{\omega}{\omega-1}$, we are left with the following result,

$$\mathcal{G}^{14\to23}(w,\bar{w})$$

$$= \delta_1(\alpha,\beta|\gamma)(w\bar{w})^{\Delta_2-2} B\left(-\Delta_1-\Delta_2+3, \frac{1}{2}(\Delta_1-\Delta_2+\Delta_3-\Delta_4+2)\right) \quad (78)$$

$$\times F_1\left(3-\Delta_1-\Delta_2, 2-\Delta_2, 2-\Delta_2, \frac{1}{2}(8+\Delta_1-\Delta_2+\Delta_3-\Delta_4), \frac{w-1}{w}, \frac{\bar{w}-1}{\bar{w}}\right).$$

---

[14]From now on we omit writing the GR contribution for each case. One should assume that they are always there inherently. The pure GR contributions impose constraints on the external conformal dimensions which is not there for quadratic gravity.

Finally using the identity of $\mathbf{F}_1$ we can write it in the following way,

$$\mathcal{G}^{14\to23}(w,\bar{w})$$

$$= \delta_1(\alpha,\beta|\gamma)B\left(-\Delta_1-\Delta_2+3,\frac{1}{2}\left(\Delta_1-\Delta_2+\Delta_3-\Delta_4+2\right)\right) \tag{79}$$

$$\times \mathbf{F}_1\left(\frac{1}{2}\left(\Delta_1-\Delta_2+\Delta_3-\Delta_4+2\right),2-\Delta_2,2-\Delta_2,\frac{1}{2}(8-\Delta_1-3\Delta_2+\Delta_3-\Delta_4),1-w,1-\bar{w}\right).$$

Finally adding the the result from these three different kinematic regimes mentioned we get the following expression,

$$\mathcal{G}^{12\to34}(w,\bar{w})+\mathcal{G}^{13\to24}(w,\bar{w})+\mathcal{G}^{14\to23}(w,\bar{w}) \tag{80}$$

$$= \delta_1(\alpha,\beta|\gamma)\Bigg[B\left(\frac{1}{2}(2+\Delta_1-\Delta_2+\Delta_3-\Delta_4),\frac{1}{2}(2+\Delta_1-\Delta_2-\Delta_3+\Delta_4)\right)$$

$$\times \mathbf{F}_1\left(\frac{1}{2}(2+\Delta_1-\Delta_2+\Delta_3-\Delta_4),2-\Delta_2,2-\Delta_2,2+\Delta_1-\Delta_2,w,\bar{w}\right)$$

$$+ (w\bar{w})^{2-\Delta_2}B(-\Delta_1-\Delta_2+3,\frac{1}{2}\left(\Delta_1-\Delta_2-\Delta_3+\Delta_4+2\right))$$

$$\times \mathbf{F}_1(3-\Delta_1-\Delta_2,2-\Delta_2,2-\Delta_2,\frac{1}{2}\left(-\Delta_1-3\Delta_2-\Delta_3+\Delta_4+8\right),\frac{1}{w},\frac{1}{\bar{w}})$$

$$+ B\left(-\Delta_1-\Delta_2+3,\frac{1}{2}\left(\Delta_1-\Delta_2+\Delta_3-\Delta_4+2\right)\right)$$

$$\times \mathbf{F}_1\left(\frac{1}{2}\left(\Delta_1-\Delta_2+\Delta_3-\Delta_4+2\right),2-\Delta_2,2-\Delta_2,\right.$$

$$\left.\frac{1}{2}(8-\Delta_1-3\Delta_2+\Delta_3-\Delta_4),1-w,1-\bar{w}\right)\Bigg].$$

Now our goal is to find the OPE coefficients using the inversion formula derived in [29]. For the inversion we need the argument of the blocks to be $w,\bar{w}$ and here we immediately identify the problem with our four-point function where we have different arguments. Hence, to resolve the problem we need the analytic continuation of the **Appell function**.

**Analytic continuation:** Here, we describe the analytic continuation of our conformal block that we found in (80). In $\mathfrak{t}$ and $\mathfrak{u}$-channel, we use analytic continuation as well as check the properties under the monodromy projection. Some portions of the analytically continued block does not contribute due to their incorrect behaviour under *monodromy projection* [28] and we discard them. For $\mathfrak{t}$ channel, we get

$$\mathbf{F}_1\left(\frac{1}{2}\left(\Delta_1-\Delta_2+\Delta_3-\Delta_4+2\right),2-\Delta_2,2-\Delta_2,\frac{1}{2}(8-\Delta_1-3\Delta_2+\Delta_3-\Delta_4),1-w,1-\bar{w}\right)$$

$$\to \frac{\Gamma(\Delta_1+\Delta_2-2)\Gamma\left(\frac{1}{2}\left(\Delta_1-\Delta_2-\Delta_3+\Delta_4+2\right)\right)}{\Gamma(\Delta_1-\Delta_2+2)\Gamma\left(\frac{1}{2}\left(\Delta_1+3\Delta_2-\Delta_3+\Delta_4-6\right)\right)}$$

$$\times \mathbf{F}_1\left(\frac{1}{2}\left(\Delta_1-\Delta_2+\Delta_3-\Delta_4+2\right),2-\Delta_2,2-\Delta_2,2-\Delta_2+\Delta_1,w,\bar{w}\right)+\ldots \tag{81}$$

In (81), the ellipsis denotes the terms obtained after analytic continuation but do not exhibit correct behaviour under monodromy projection. The analytic continuation for the $\mathfrak{u}$-channel kinematics can be done similarly. Now we proceed to compute the OPE coefficients eventually in the $\mathfrak{s}$-channel kinematics. This choice is generic due to choosing external primary conformal dimension(s) to be equal. Otherwise, one should calculate the OPE coefficients for different channels separately, in euclidean setup. Though in euclidean scenario there is no sense of time, and therefore operator ordering does not matter.

### 4.3 Conformal block expansion and partial wave coefficients

For $12 \to 34$ kinematics, the four point function of the conformal primaries (with one shadowed operator) can be expanded in s-channel OPE in the following way [29, 63],

$$\mathcal{G}^{12\to34}(w,\bar{w}) = \sum_{J,\Delta} f_{12\mathcal{O}} f_{34\mathcal{O}} G_{J,\Delta}(w,\bar{w}), \tag{82}$$

where, $J, \Delta$ are the spin and conformal dimension of the exchanging primary operator ($\mathcal{O}$). $f_{ijk}$ are the OPE coefficients and $G_{J,\Delta}(z,\bar{z})$ is defined as

$$G_{J,\Delta} = \frac{k_{\Delta-J}(w)k_{\Delta+J}(\bar{w}) + k_{\Delta+J}(w)k_{\Delta-J}(\bar{w})}{1+\delta_{J,0}}, \quad \text{with } k_{\beta}(w) = w^{\beta/2} {}_2F_1\left(\beta/2+a, \beta/2+b, \beta, w\right), \tag{83}$$

where the constants $a, b$ are identified as $a = \frac{1}{2}(2-\Delta_2-\Delta_1), b = \frac{1}{2}(\Delta_3-\Delta_4)$. Therefore the block should take the form [14, 29]

$$\begin{aligned}
\mathcal{G}^{12\to34}(w,\bar{w}) &= \delta_1(\alpha,\beta|\gamma)B\left(\frac{1}{2}(2+\Delta_1-\Delta_2+\Delta_3-\Delta_4), \frac{1}{2}(2+\Delta_1-\Delta_2-\Delta_3+\Delta_4)\right) \\
&\quad \sum_{J,\Delta} f_{12\mathcal{O}} f_{34\mathcal{O}}\left[w^{\frac{\Delta-J}{2}}\bar{w}^{\frac{\Delta+J}{2}}\right] \\
&\quad \times {}_2F_1\left(\begin{matrix}\frac{1}{2}(\Delta-J+2-\Delta_2-\Delta_1), \frac{1}{2}(\Delta-J+\Delta_3-\Delta_4)\\ \Delta-J\end{matrix}; w\right) \\
&\quad \times {}_2F_1\left(\begin{matrix}\frac{1}{2}(\Delta+J+2-\Delta_2-\Delta_1), \frac{1}{2}(\Delta+J+\Delta_3-\Delta_4)\\ \Delta+J\end{matrix}; \bar{w}\right) + (J\to-J).
\end{aligned} \tag{84}$$

Next, our goal is to compute the OPE coefficients using two different approaches: Burchnall-Chaundy expansion and OPE inversion formula. (Un-)fortunately the first one is only applicable for the four point functions involving the Appell functions.

### Burchnall-Chaundy expansion:

We use the Burchnall-Chaundy expansion[15] [66] (see [65] for more details) of the Appell function in (75) to get the following,

$$\begin{aligned}
&\mathcal{G}^{12\to34}(w,\bar{w}) \tag{86}\\
&= \delta_1(\alpha,\beta|\gamma)B\left(\frac{1}{2}(2+\Delta_1-\Delta_2+\Delta_3-\Delta_4), \frac{1}{2}(2+\Delta_1-\Delta_2-\Delta_3+\Delta_4)\right) \\
&\quad \times w^{\frac{2+\Delta_1-\Delta_2}{2}}\bar{w}^{\frac{2+\Delta_1-\Delta_2}{2}} \\
&\quad \sum_{n=0}^{\infty} \frac{\left(\frac{1}{2}(2+\Delta_1-\Delta_2+\Delta_3-\Delta_4)\right)_n (2-\Delta_2)_n(2-\Delta_2)_n \left(\frac{1}{2}(2+\Delta_1-\Delta_2-\Delta_3+\Delta_4)\right)_n}{n!(1+n-\Delta_2+\Delta_1)_n(2+\Delta_1-\Delta_2)_{2n}} \\
&\quad \times w^n\bar{w}^n {}_2F_1\left(\begin{matrix}\frac{1}{2}(2+\Delta_1-\Delta_2+\Delta_3-\Delta_4)+n, 2-\Delta_2+n\\ \Delta_1-\Delta_2+2+2n\end{matrix}; w\right) \\
&\quad \times {}_2F_1\left(\begin{matrix}\frac{1}{2}(2+\Delta_1-\Delta_2+\Delta_3-\Delta_4)+n, 2-\Delta_2+n\\ \Delta_1-\Delta_2+2+2n\end{matrix}; \bar{w}\right).
\end{aligned}$$

---

[15]The *Burchnall-Chaundy expansion* of the Appell hypergeometric function enables one to write it in terms of a product of two Gauss hypergeometric functions in the following way,

$$\mathbf{F}_1(a, b_1, b_2, c, x, y) = \sum_{n=0}^{\infty} \frac{(a)_n(b_1)_n(b_2)_n(c-a)_n}{n!(c+n-1)_n(c)_{2n}} x^n y^n \times {}_2F_1\left(\begin{matrix}a+n, b_1+n\\ c+2n\end{matrix}; x\right){}_2F_1\left(\begin{matrix}a+n, b_2+n\\ c+2n\end{matrix}; y\right). \tag{85}$$

Now comparing (84) with (86), we get

$$\Delta \pm J = 2 + 2n + \Delta_1 - \Delta_2.$$ (87)

This is only possible when $J = 0$ and immediately implies that the exchange operators have to be scalars. Therefore the $\text{OPE}_{12\mathcal{O}} \otimes \text{OPE}_{34\mathcal{O}}$ coefficient (for $J = 0$) is given by

$$f_{12\mathcal{O}}f_{34\mathcal{O}}$$ (88)
$$\sim \delta_1(\alpha,\beta|\gamma)B\left(\frac{1}{2}(2+\Delta_1-\Delta_2+\Delta_3-\Delta_4), \frac{1}{2}(2+\Delta_1-\Delta_2-\Delta_3+\Delta_4)\right)$$
$$\times \frac{\left(\frac{1}{2}(2+\Delta_1-\Delta_2+\Delta_3-\Delta_4)\right)_n(2-\Delta_2)_n(2-\Delta_2)_n\left(\frac{1}{2}(2+\Delta_1-\Delta_2-\Delta_3+\Delta_4)\right)_n}{n!(1+n-\Delta_2+\Delta_1)_n(2+\Delta_1-\Delta_2)_{2n}},$$

with $2n = \Delta - 2 - \Delta_1 + \Delta_2$. For simplicity we set the conformal dimensions of the external primaries to be same $\Delta_{\mathcal{O}}$. Therefore the OPE coefficient reduces to (for non-spinning exchange)

> **OPE coefficient for EFT correction (scalar exchange)**
>
> $$f_{\mathcal{O}\mathcal{O}\mathcal{O}}^2 = \delta_1(\alpha,\beta|\gamma \to 4(\Delta_{\mathcal{O}}-1))B(\Delta_{\mathcal{O}},\Delta_{\mathcal{O}}) \times \frac{\Gamma\left(\frac{\Delta}{2}\right)^4\Gamma(2\Delta_{\mathcal{O}})\Gamma\left(\frac{\Delta}{2}+\Delta_{\mathcal{O}}-1\right)}{\Gamma(\Delta-1)\Gamma(\Delta)\Gamma\left(\frac{\Delta}{2}-\Delta_{\mathcal{O}}+1\right)\Gamma(\Delta_{\mathcal{O}})^4}.$$
> (89)

This naturally leads us to the question: what happens if we consider the exchange of spinning primaries, i.e., operators with non-zero spin ($J \neq 0$)?[16] To address this question, we make use of Caron-Huot's *OPE inversion formula* [29], which applies for operators with finite, non-zero spin subject to the appropriate unitarity bounds. *If one can demonstrate that the four-point function admits a consistent inversion yielding non-vanishing OPE coefficients for spinning operators, this would imply that the spectrum necessarily includes spinning exchanges. Thus, the problem reduces to examining whether such a consistent inversion is possible. We show that, in this case, the OPE inversion can indeed be performed consistently. Furthermore, we conduct a comparative study of the results obtained for $J = 0$ using both approaches.*

**OPE inversion:**

To start with, one needs the following integral representation. The discreteness in $\Delta$ should be converted into an integral form, sometimes known as partial-wave expansion [63,67] and takes the following form [29],

$$\mathcal{G}^{12\to34}(w,\bar{w}) = \mathbf{1}_{12}\mathbf{1}_{34} + \sum_{J=0}^{\infty}\int_{d/2-i\infty}^{d/2+i\infty}\frac{d\Delta}{2\pi i}\,c(J,\Delta)F_{J,\Delta}(w,\bar{w}),$$ (90)

where, $c(J,\Delta)$ is the partial-wave coefficient and $F_{J,\Delta}$ is given by

$$F_{J,\Delta}(w,\bar{w}) = \frac{1}{2}\left(G_{J,\Delta}(w,\bar{w}) + \underbrace{\frac{K_{J,d-\Delta}}{K_{J,\Delta}}G_{J,d-\Delta}(w,\bar{w})}_{\text{Shadow contribution}}\right),$$ (91)

---

[16]It is not entirely clear to us why the Burchnall-Chaundy expansion of the four-point function fails to capture the complete spectrum of the theory.

where the constants can be casted as

$$K_{J,\Delta} = \frac{\Gamma(\Delta-1)}{\Gamma(\Delta-d/2)}\kappa_{J+\Delta}, \qquad \kappa_\beta = \frac{\Gamma(\beta/2-a)\Gamma(\beta/2+a)\Gamma(\beta/2-b)\Gamma(\beta/2+b)}{2\pi^2\Gamma(\beta-1)\Gamma(\beta)},$$
$$a = \frac{1}{2}(2-\Delta_2-\Delta_1), \qquad b = \frac{1}{2}(\Delta_3-\Delta_4), \tag{92}$$

and $w, \bar{w}$ are the cross-ratio(s). $\Delta_i$, for $i = 1,\dots,4$ are conformal dimension of primaries.[17] Under the assumption that harmonic functions $F(J,\Delta)$ are orthogonal to each other, and Euclidean OPE data(s) can be obtained by inverting (90) in the following way [29],

$$c_s(J,\Delta) = N(J,\Delta)\int_{-\infty}^{\infty} d^2w\,\mu(w,\bar{w})\,F_{J,\Delta}(w,\bar{w})\,\mathcal{G}^{12\to34}(w,\bar{w}), \tag{93}$$

where

$$\mu(w,\bar{w}) = \left|\frac{w-\bar{w}}{w\bar{w}}\right|^{d-2}\frac{(1-w)^{a+b}(1-\bar{w})^{a+b}}{(w\bar{w})^2}, \qquad \text{with} \qquad a = \frac{2-\Delta_2-\Delta_1}{2}, \quad b = \frac{\Delta_3-\Delta_4}{2},$$
$$G_{J,\Delta} = \frac{k_{\Delta-J}(w)k_{\Delta+J}(\bar{w})+k_{\Delta+J}(w)k_{\Delta-J}(\bar{w})}{1+\delta_{J,0}}, \quad \text{with} \quad k_\beta(w) = w^{\beta/2}\,{}_2\mathbf{F}_1\left(\beta/2+a,\beta/2+b,\beta,w\right). \tag{94}$$

Now, we can decompose the integral (93) into three different channels, and as we are interested in the OPE limit, we make the following variable change

$$w = \frac{4\rho_w}{(1+\rho_w)^2}, \qquad \bar{w} = \frac{4\rho_{\bar{w}}}{(1+\rho_{\bar{w}})^2}, \qquad |\rho_w| < 1,$$

and focus on the $(0,1)$ region as we are interested in the s-channel OPE. According to the chosen notion of cross ratio, the s-channel OPE in Euclidean case dominates in this specific regime. So finally, we get[18]

$$c_s(J,\Delta) \tag{95}$$
$$= N(J,\Delta)\int_0^1\int_0^1 d\rho_w d\rho_{\bar{w}}\,\mu(\rho_w,\rho_{\bar{w}})F_{J,\Delta}(\rho_w,\rho_{\bar{w}})\mathcal{G}^s(\rho_w,\rho_{\bar{w}})$$
$$= \delta_1(a,\beta|\gamma)N(J,\Delta)\int_{|\rho_w|\ll 1} d^2\rho_w\,\mu(\rho_w,\rho_{\bar{w}})\left(\left(\frac{4\rho_w}{(1+\rho_w)^2}\right)^{\frac{\Delta-J}{2}}\left(\frac{4\rho_{\bar{w}}}{(1+\rho_{\bar{w}})^2}\right)^{\frac{\Delta+J}{2}}+\rho_w\leftrightarrow\rho_{\bar{w}}\right)$$
$$\times \mathbf{F}_1\left(\Delta_\mathcal{O},\Delta_\mathcal{O},2\Delta_\mathcal{O},\frac{4\rho_w}{(1+\rho_w)^2},\frac{4\rho_{\bar{w}}}{(1+\rho_{\bar{w}})^2}\right)$$
$$\approx \delta_1(a,\beta|\gamma)N(J,\Delta)\int_0^1\int_0^1 \frac{d\rho_w d\rho_{\bar{w}}}{16\rho_w^2\rho_{\bar{w}}^2}\left(\rho_w^{\frac{\Delta-J}{2}}\rho_{\bar{w}}^{\frac{\Delta+J}{2}}+\rho_w^{\frac{\Delta-J}{2}}\rho_{\bar{w}}^{\frac{\Delta+J}{2}}\right)\left(\frac{(\rho_{\bar{w}}+1)^2}{(\rho_{\bar{w}}-1)^2}\right)^{\Delta_\mathcal{O}}\left(1-\rho_w^2\right)\left(1-\rho_{\bar{w}}^2\right)$$
$$= \delta_1(a,\beta|\gamma)\frac{e^{-2\pi i\Delta_\mathcal{O}}N(J,\Delta)}{8(\Delta+J-2)}\Gamma(1-2\Delta_\mathcal{O})\left[\Gamma\left(\frac{1}{2}(-J+\Delta-2)\right)\right.$$
$$\left.\times {}_2\tilde{F}_1\left(\frac{1}{2}(-J+\Delta-2),-2\Delta_\mathcal{O};\frac{1}{2}(-J+\Delta-4\Delta_\mathcal{O});-1\right)+J\leftrightarrow -J\right],$$
$$\text{with } \mathcal{R}[\Delta-J] > 2, \quad \text{and} \quad \mathcal{R}[\Delta+J] > 2.$$

---

[17]Note that for our case we have three primaries and one shadowed primary.

[18]We have used the relation $k_\beta(w,\bar{w})\Big|_{a=0,b=0} = (4\rho)^{\beta/2}\,{}_2F_1(\frac{1}{2},\frac{\beta}{2},\frac{\beta+1}{2},\rho^2)$.

In evaluating (95), the integral convergence condition of shadow part of the conformal block blacks unitarity bound, hence can be dropped. In simplifying the integral at the Euclidean OPE limit ($\rho_w, \rho_{\bar{w}} \ll 1$), we have approximated the hypergeometric function to be 1. Similarly we have approximated the whole Appell $\mathbf{F}_1$ function as[19]

$$\mathbf{F}_1\left(\Delta_{\mathcal{O}}, \Delta_{\mathcal{O}}, \Delta_{\mathcal{O}}, 2\Delta_{\mathcal{O}}, \frac{4\rho_w}{(1+\rho_w)^2}, \frac{4\rho_{\bar{w}}}{(1+\rho_{\bar{w}})^2}\right) \approx \left(\frac{\rho_{\bar{w}}+1}{\rho_{\bar{w}}-1}\right)^{2\Delta_{\mathcal{O}}}. \tag{96}$$

Now, one can extract the OPE coefficient from the partial-wave coefficient using the following observation that $c(J, \Delta)$ has poles at the real $\Delta$ axis at the location of the physical operators and consequently by computing the residues [29, 68],

$$c(J, \Delta') \sim -\sum_{\Delta} \frac{f^2_{\mathcal{O}\mathcal{O}\mathcal{O}_{\Delta}}}{\Delta' - \Delta}. \tag{97}$$

In the complex $\Delta'$-plane, the integrand has several poles originating from various $\Gamma$-functions and potentially from the hypergeometric function as well. By analyzing (95), it becomes evident that, to ensure proper convergence of the integral, the contour must be closed on the right side of the $\Delta'$-plane i.e. $\mathrm{Re}(\Delta') > 1$. If we consider the shadow contribution of the block, the contour must be closed on the left. However, due to shadow symmetry, the OPE coefficients remain unchanged, as discussed in [29]. Accordingly, we evaluate the residues at the poles that lie in the right half of the complex $\Delta'$-plane. We now turn to the function of interest (95):

$$c_{\mathfrak{s}}(J, \Delta) \tag{98}$$

$$= \delta_1(\alpha, \beta | \gamma) B(\Delta_{\mathcal{O}}, \Delta_{\mathcal{O}}) \Gamma(1 - 2\Delta_{\mathcal{O}}) \frac{\Gamma^4\left(\frac{\Delta'+J}{2}\right) \Gamma(2 - \Delta' + J - 1) \Gamma(2 - \Delta' + J)}{2\pi(\Delta' + J - 2)\Gamma(\Delta' + J - 1)\Gamma(\Delta' + J)\Gamma^4\left(\frac{2-\Delta'+J}{2}\right)}$$

$$\times \left[\Gamma\left(\frac{1}{2}(\Delta' - J - 2)\right) {}_2\tilde{F}_1\left(\frac{1}{2}(-J + \Delta' - 2), -2\Delta_{\mathcal{O}}; \frac{1}{2}(-J + \Delta' - 4\Delta_{\mathcal{O}}); -1\right) + J \leftrightarrow -J\right].$$

We analyze the singularities of $c_{\mathfrak{s}}(J, \Delta)$ piece by piece. The function has three types of simple poles for $\mathrm{Re}(\Delta') > 1$:[20]

- An infinite tower of simple poles at $\Delta' = n + J + 1$,

- Another infinite tower of simple poles at $\Delta' = n + J + 2$,

- There is a single simple pole at $\Delta' = 2 - J$. However, based on the convergence condition of the integral in (95), this pole lies outside the region of convergence and can therefore be excluded from the residue analysis.

---

[19]The normalization factor $N(J, \Delta)$ in general dimension is given by,

$$N(J, \Delta) = \frac{4^{\Delta} \Gamma(J + \frac{d-2}{2})\Gamma(J + \frac{d}{2})K_{J,\Delta}}{2\pi \Gamma(J+1)\Gamma(J+d-2)K_{J,d-\Delta}} B\left(\frac{1}{2}(2 + \Delta_1 - \Delta_2 + \Delta_3 - \Delta_4), \frac{1}{2}(2 + \Delta_1 - \Delta_2 - \Delta_3 + \Delta_4)\right).$$

[20]One can verify that ${}_2\tilde{F}_1$ does not introduce any poles for $\mathrm{Re}\,\Delta' > 1$ by observing that, for $\Delta_{\mathcal{O}} = 1$, it behaves as, ${}_2\tilde{F}_1 \sim -\frac{2(-\Delta+J+2)}{\Gamma(\frac{1}{2}(-J+\Delta-2))}$, which is manifestly analytic in this region. Upon analytically continuing $\Delta_{\mathcal{O}}$ into the entire complex plane while keeping its real part fixed, no additional pole structure is expected to emerge.

The corresponding residues are:

- at $\Delta' = n + J + 2$:

$$\operatorname*{Res}_{\Delta'=n+J+2} c_{\mathsf{s}}(J,\Delta') \tag{99}$$

$$= \frac{e^{-2i\pi\Delta_{\mathcal{O}}}\delta_1(\alpha,\beta|\gamma)B(\Delta_{\mathcal{O}},\Delta_{\mathcal{O}})\Gamma(1-2\Delta_{\mathcal{O}})2^{2J+2n}\Gamma\left(J+\frac{n}{2}+1\right)^4}{n^3(2J+n)^3\Gamma\left(-\frac{n}{2}\right)^4\Gamma(n)\Gamma(n+2)\Gamma(2J+n)\Gamma(2J+n+2)\Gamma\left(\frac{n}{2}-2\Delta_{\mathcal{O}}+1\right)\Gamma\left(J+\frac{n}{2}-2\Delta_{\mathcal{O}}+1\right)}$$

$$\times\left[-2n(2J+n)\Gamma\left(J+\frac{n}{2}+1\right)\Gamma\left(\frac{n}{2}-2\Delta_{\mathcal{O}}+1\right)\frac{ab}{c^2}{}_2\Theta_1^{(1)}\left(\begin{array}{c}1,1:c,a+1;b+1\\c+1:2;c+1\end{array};-1,-1\right)\right.$$

$$+2n(2J+n)\left(-\Gamma\left(\frac{n}{2}+1\right)\left(\frac{ab}{c^2}{}_2\Theta_1^{(1)}\left(\begin{array}{c}1,1:c,a+1;b+1\\c+1:2;c+1\end{array};-1,-1\right)\right.\right.$$

$$-\frac{a}{c}\Theta_1^{(1)}\left(\begin{array}{c}1,1:a,a+1;b+1\\a+1:2;c+1\end{array};-1,-1\right)\right)$$

$$\times\Gamma\left(J+\frac{n}{2}-2\Delta_{\mathcal{O}}+1\right)-\frac{a}{c}\Gamma\left(J+\frac{n}{2}+1\right)\Gamma\left(\frac{n}{2}+2\Delta_{\mathcal{O}}+1\right)$$

$$\times\Theta_1^{(1)}\left(\begin{array}{c}1,1:a,a+1;b+1\\a+1:2;c+1\end{array};-1,-1\right)\right)$$

$$+n\left(2\Gamma\left(J+\frac{n}{2}+1\right)\Gamma\left(\frac{n}{2}-2\Delta_{\mathcal{O}}+1\right){}_2F_1\left(J+\frac{n}{2},-2\Delta_{\mathcal{O}};J+\frac{n}{2}-2\Delta_{\mathcal{O}}+1;-1\right)\right.$$

$$\times\left(\left(2J+n\right)\left(H_{J+\frac{n}{2}-2\Delta_{\mathcal{O}}}-5H_{J+\frac{n}{2}-1}+4H_{2J+n}-4H_{-\frac{n}{2}-1}+4H_n-4\log(2)\right)\right.$$

$$\left.+\frac{4J-2}{n+1}-\frac{2}{2J+n+1}-4\right)$$

$$+n\Gamma\left(\frac{n}{2}\right){}_2F_1\left(\frac{n}{2},-2\Delta_{\mathcal{O}};\frac{1}{2}(n-4\Delta_{\mathcal{O}}+2);-1\right)\Gamma\left(J+\frac{n}{2}-2\Delta_{\mathcal{O}}+1\right)$$

$$\times\left(-(2J+n)\left(4H_{J+\frac{n}{2}-1}-4H_{2J+n}-H_{\frac{n}{2}-2\Delta_{\mathcal{O}}}+4H_{-\frac{n}{2}-1}-4H_n+\psi^{(0)}\left(\frac{n}{2}\right)+\gamma_E\right)\right.$$

$$\left.+\frac{4J-2}{n+1}-\frac{2}{2J+n+1}-4\log(2)(2J+n)-2\right)\right)$$

$$+4(2J+n)\Gamma\left(J+\frac{n}{2}+1\right)\Gamma\left(\frac{n}{2}-2\Delta_{\mathcal{O}}+1\right){}_2F_1\left(J+\frac{n}{2},-2\Delta_{\mathcal{O}};J+\frac{n}{2}-2\Delta_{\mathcal{O}}+1;-1\right)\left.\right].$$

- at $\Delta' = n + J + 1$:

$$\operatorname*{Res}_{\Delta'=n+J+1} c_{\mathsf{s}}(J,\Delta') \tag{100}$$

$$= \delta_1(\alpha,\beta|\gamma)B(\Delta_{\mathcal{O}},\Delta_{\mathcal{O}})\frac{1}{(n-1)n!(2J+n-1)\Gamma\left(\frac{1-n}{2}\right)^4\Gamma(2J+n)\Gamma(2J+n+1)}$$

$$\times\left[e^{-2i\pi\Delta_{\mathcal{O}}}(-1)^n\Gamma(1-2\Delta_{\mathcal{O}})4^{J+n}\Gamma(1-n)\Gamma\left(J+\frac{n}{2}+\frac{1}{2}\right)^4\right.$$

$$\times\left(\Gamma\left(J+\frac{n}{2}+\frac{1}{2}\right){}_2\tilde{F}_1\left(\frac{1}{2}(2J+n-1),-2\Delta_{\mathcal{O}};\frac{1}{2}(2J+n-4\Delta_{\mathcal{O}}+1);-1\right)\right.$$

$$\left.\left.+\Gamma\left(\frac{n+1}{2}\right){}_2\tilde{F}_1\left(\frac{n-1}{2},-2\Delta_{\mathcal{O}};\frac{1}{2}(n-4\Delta_{\mathcal{O}}+1);-1\right)\right)\right].$$

Here, $H_n = \int_0^1 dx \frac{1-x^n}{1-x}$ denotes the harmonic number, and $\psi^{(0)}$ is the digamma function, which is the logarithmic derivative of the gamma function. Moreover, we write the derivatives of the hypergeometric functions (appeared in the calculation) can be casted in terms of *Kampé*

*de Fériet-like functions* [69].[21]

$$_2F_1^{(1,0,0,0)}(a,b,c,-1) = -\frac{a}{c}\Theta_1^{(1)}\left(\begin{array}{c} 1,1:a,a+1;b+1 \\ a+1:2;c+1 \end{array};-1,-1\right),$$

$$_2F_1^{(0,0,1,0)}(a,b,c,-1) = \frac{ab}{c^2}{}_2\Theta_1^{(1)}\left(\begin{array}{c} 1,1:c,a+1;b+1 \\ c+1:2;c+1 \end{array};-1,-1\right), \qquad (101)$$

$$\text{with } a = \frac{n}{2}, \quad b = -2\Delta_{\mathcal{O}}, \quad c = \frac{n}{2} - 2\Delta_{\mathcal{O}} + 1.$$

Collecting all the results from (99) and (100) we find the OPE coefficient to be

---

**OPE coefficient for EFT correction (general exchange)**

$$f^2_{\Delta_{\mathcal{O}}\Delta_{\mathcal{O}}\Delta} = -\operatorname*{Res}_{\Delta'=n+J+2} c_{\mathsf{s}}(J,\Delta')\bigg|_{n=\Delta-J-2} - \operatorname*{Res}_{\Delta'=n+J+1} c_{\mathsf{s}}(J,\Delta')\bigg|_{n=\Delta-J-1}.$$

---

*A comparison of OPE computed from BC expansion and OPE inversion formula*: Now we present a comparative analysis of the s-channel OPE, which offers valuable insights into the interactions among conformal primary operators on the celestial sphere. In Fig. (5), we present a comparative study of extracting the OPE coefficients (taking the external conformal dimension to be $\Delta_{\mathcal{O}} = 1 + i$) from the two approaches: Burchnall-Chaundy (BC) expansion and OPE inversion. While the OPE coefficients derived via the BC expansion show agreement with those obtained through the OPE inversion formula for $J = 0$ and in the vicinity of: $\text{Im}(\Delta_{\mathcal{O}}) \gtrsim 1$, discrepancies begin to emerge at higher values of $\text{Im}(\Delta_{\mathcal{O}})$, leaving no major structural differences. These deviations can be attributed to the limit $\rho_w \ll 1$, which approximates the Appell and hypergeometric functions encoding the $\Delta_{\mathcal{O}}$ dependence as shown in (96). Due to the technical complexity of performing an exact OPE inversion without such approximations, we currently work within a simplified framework. However, it is anticipated that a more complete evaluation of the OPE inversion, incorporating the full integration domain, may yield results that are more consistent with those from the BC expansion. The exact evaluation of OPE coefficients using the inversion formula requires a separate study, which we leave for further investigation.

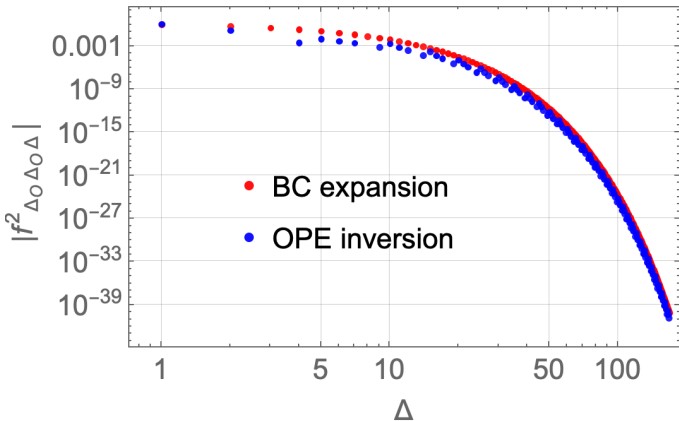

Figure 5: Plot depicting the matching for extraction of OPE coefficient using BC expansion (**red**) and OPE inversion for spin-0 (**blue**). We have set the conformal dimension for the external primaries to be: $\Delta_{\mathcal{O}} = 1 + i$.

---

[21]Specifically, $_2\Theta_1^{(1)}\left(\begin{array}{c} a_1,a_2 : b_1,b_2;b_3 \\ c_1 : d_1;d_2 \end{array};x_1,x_2\right) = \sum_{m_1=0}^{\infty}\sum_{m_2=0}^{\infty}\frac{(a_1)_{m_1}(a_2)_{m_2}(b_1)_{m_1}(b_2)_{m_1+m_2}(b_3)_{m_1+m_2}}{(c_1)_{m_1}(d_1)_{m_1+m_2}(d_2)_{m_1+m_2}}\frac{x_1^{m_1}x_2^{m_2}}{m_1!m_2!}.$

This calculation demonstrates a way for extracting OPE data from the four-point function within a suitable approximation scheme. Therefore, the answer to the third question raised in the introduction is also affirmative. Once analytic control over the OPE data is established for the Born amplitude, we immediately gain control over the full non-perturbative eikonal celestial amplitude, at least for the EFT correction part, since the $\omega$-integral in (66) is independent of the cross-ratio $z$. Consequently, the same analysis performed for the Born amplitude case can be applied, with the $\omega$-integral treated as a formal object. However, for the GR part in (67), the functional dependence of the shadow amplitude on $z$ is highly sensitive to the $\omega$-integral, for which no closed-form solution is available. As a result, we lose entirely the analytic control over the OPE data for the GR part of the eikonal shadow correlator.

## 5 Connection to Carrollian amplitude

A significant bridge between celestial and Carrollian frameworks is established through the so-called $\mathcal{B}$-transform [70, 71]. This transformation maps celestial amplitudes defined via Mellin transforms of scattering amplitudes in momentum space into Carrollian amplitudes, which are naturally formulated in a spacetime with Carrollian symmetry arising in the ultra-relativistic limit. The $\mathcal{B}$-transform acts as a change of basis, translating between representations adapted to conformal structures at null infinity and those suited to Carrollian dynamics. Notably, the transformation preserves the essential symmetry content of the theory, shedding light on the interplay between conformal covariance on the celestial sphere and Carrollian symmetries on null hypersurfaces [70–72]. Now we can compute the unmodified Carrollian amplitude from the celestial amplitude as

$$\mathcal{C}(u_i, z_i, \bar{z}_i) \sim \int_0^\infty \prod_i d\omega_i \, e^{-i \sum_j \epsilon_j \omega_j u_j} \mathbb{M}_{\text{eik}}(\omega_i, z_i) \tag{102}$$

$$\sim \prod_{i=1}^4 \int_0^1 d\sigma_i \delta\left(\sum_i \sigma_i - 1\right)[\cdots] \int_0^\infty d\nu \, \nu^{-1} e^{-i\nu^{-1}(\sigma_1 u_1 + \sigma_2 u_2 - \sigma_3 u_3 - \sigma_4 u_4)} \mathbb{M}_{eik}(\nu, z)$$

$$\sim \prod_{i=1}^4 \int_0^1 d\sigma_i \delta\left(\sum_i \sigma_i - 1\right)[\cdots]\left(\int_0^\infty d\nu \, \nu^{-1} e^{-i\nu^{-1}(\sigma_1 u_1 + \sigma_2 u_2 - \sigma_3 u_3 - \sigma_4 u_4)}\right.$$

$$\left.\times \left[\frac{1}{48\kappa}\nu^2\left(\frac{\kappa\left(3\kappa - \nu^2(-\alpha + 8\beta)\right)}{(\kappa - \alpha\nu^2)(\kappa - 2\beta\nu^2)} - \frac{4\kappa}{\kappa + \alpha\nu^2} + \frac{\kappa}{\kappa + 2\beta\nu^2} - \frac{24z}{(z-1)}\right)\right]\right).$$

In obtaining second line from the first one we used (8). Now, changing the variable $\nu^{-1} = p$ and $(\sigma_1 u_1 + \sigma_2 u_2 - \sigma_3 u_3 - \sigma_4 u_4) = h$, we can write the integrand in the parenthesis as

$$\tilde{\mathcal{C}}(\sigma_i, z_i, \bar{z}_i)$$

$$= \frac{h^8 z}{22579200\sqrt{2\pi}\,\kappa(z-1)}\left(-761 + 280\left(-\frac{1}{2}(i\pi)\,\text{sign}(h) + \log(h) + \gamma_E\right)\right) \tag{103}$$

$$-(-\beta + 2\alpha)\frac{h^{10}}{109734912000\sqrt{2\pi}\,\kappa^2}\left(-7381 + 2520\left(-\frac{1}{2}(i\pi)\,\text{sign}(h) + \log(h) + \gamma_E\right)\right),$$

where $\gamma_E$ is the Euler's constant. The first and second terms in (103) come from the GR and the first-order correction to GR in quadratic EFT, respectively. Now, performing the $\sigma$ integral, one can easily find out the Carrollian amplitude corresponding to the celestial amplitude as,

$$\mathcal{C}(u_i, z_i, \bar{z}_i) \sim \prod_{i=1}^4 \int_0^1 d\sigma_i \delta(\sigma_i - \sigma_{*i})\tilde{\mathcal{C}}(\sigma_i, z_i, \bar{z}_i), \tag{104}$$

where the localization point $\sigma_{i\star}$ can be found in (A.6).

One important point to note here is that the Carrollian amplitude has an IR pole in GR, which now shifts due to the non-zero value of $\alpha, \beta$. However, we have found this result by linearizing in $\alpha, \beta$. The IR-pole behaviour is given by

$$\lim_{\delta \to 0^+} \frac{1}{\delta} \frac{h^8(z-1)\left(h^{2\frac{(z-1)}{z}}(-\beta + 2\alpha) + 540\kappa\right)}{43545600\sqrt{2\pi}\kappa z}. \tag{105}$$

Here, the Infrared pole is important for boost invariance [71].

# 6 Conclusion and discussion

Motivated by the case study of celestial eikonal amplitudes and their improved analytic behaviour, we have generalized the study of it for Einstein gravity to quadratic EFT. Below, we list the main findings of our paper,

1. Motivated by the analysis of [41], we construct the celestial eikonal amplitude for the quadratic EFT of gravity, despite the fact that the Born amplitude in this case is meromorphic, unlike in GR. We find that the corrections to the eikonal phase arising from the EFT are short-ranged, involving $\delta$-function contributions. Nevertheless, by adopting a suitable prescription for handling functions of the $\delta$-function, we demonstrate that it is possible to extract physically meaningful results even in the presence of such contact interactions. We find that like GR the eikonal amplitude is multiplication of two parts: Born amplitude and a phase.

2. Furthermore, we analyze the analytic structure of the eikonal amplitude by examining its behavior in both the ultrablack (UV) and infrared (IR) regimes. In the UV limit, we find that the amplitude exhibits no structural differences compared to GR. However, upon numerical checking we find the EFT correction part converges faster than the GR part. But in the IR regime, the leading singularity is identified as a simple pole, in contrast to GR, where the leading singularity corresponds to an $n$-th order pole. Thus, the infrared behavior appears to improve upon the inclusion of EFT corrections. Although we are unable to compute the $\omega$ integral exactly, we derive the corresponding dispersion relation. This dispersion relation is modified due to the presence of non-vanishing coupling constants $(\alpha, \beta)$. Contributions from poles on the real axis are absent, as they are excluded by the choice of integration contour, which is essential to ensure that the limit $(\alpha, \beta) \to 0$ correctly.

3. We also compute the celestial operator product expansion (OPE) from a four-point function involving three primary operators and one shadow operator, by expressing the correlator in the basis of *conformal primary wavefunctions*. Upon evaluating the shadowed four-point function, we obtain an Appell function, which we then decompose into a sum of products of hypergeometric functions using the Burchnall-Chaundy expansion. In this process, we identify the celestial conformal blocks and express them in terms of hypergeometric functions. Remarkably, the underlying symmetries fix the conformal dimensions of the exchanged operators in a manner consistent with the Osborn block expansion, allowing the remaining factors to be identified as OPE coefficients. While such coefficients can alternatively be extracted from the collinear limit of scattering amplitudes, yielding only the leading OPE behaviour; our computation proceeds without invoking this limit. Surprisingly, OPE coefficient obtained using Burchnall-Chaundy does not involve contribution from the spinning exchange. To get the contribution from spin, we employ

the (Euclidean) OPE inversion formula to extract the full set of OPE coefficients. Furthermore, we provide a comparison of OPE coefficients extracted from the two ways mentioned above.

Now, we end this section by discussing some possible future outlooks. The first objective is to compute the OPE inversion exactly, without resorting to approximations, for both the GR component and the EFT-corrected component. This task presents a significant challenge, particularly for the GR contribution, as it requires the exact computation of the eikonal amplitude. While the EFT corrections generally allow for a more straightforward (as the kinematic part factors out form the eikonal amplitude) inversion, the GR component remains nontrivial due to the inherent complexity of exact eikonal amplitude calculations in this context. Moreover, due to the infrared triangle between soft theorems, ward identities, and memory effects, as proposed by Strominger et al. [3, 73], it is an open arena for investigating ward identities in quadratic EFT. One can try to find the memory effects in this setup. One can also attempt to compute soft theorems and central charges in this case. In effective field theories (EFTs), gravitons typically possess at least one massive mode. Consequently, when attempting to construct the stress-energy tensor within the celestial framework, one must invoke the shadow transform of the massless graviton mode. However, the presence of residual massive degrees of freedom complicates this process, rendering the stress tensor's construction nontrivial and subtle. Last but not the least, One can also consider light transforms in celestial CFT for the quadratic gravity and focus on marginal operator construction [74–76].

# Acknowledgments

The authors would like to thank Wei Fan for useful email correspondence. We also thank Mousumi Maitra for collaborating at the initial stage of the project. AB would like to thank the Department of Physics of BITS Pilani, Goa Campus, for hospitality during the course of this work, as well as the organizers of the "Holography, strings and other fun things II" workshop there.

**Funding information**  S.G (PMRF ID: 1702711) and S.P (PMRF ID: 1703278) are supported by the Prime Minister's Research Fellowship of the Government of India. S.G and S.P would like to thank the "Strings 2025" organizing committee for giving the opportunity to present posters and NYUAD (New York University Abu Dhabi) for their kind hospitality during the course of the work. AB is supported by the Core Research Grants (CRG/2023/ 001120 and CRG/2023/005112) by DST-ANRF of India Govt. AB also acknowledges the associateship program of the Indian Academy of Science, Bengaluru.

# A  Few definitions and conventions

- Hypergeometric $_2\mathbf{F}_1$ satisfies the following identities,

$$
\begin{aligned}
{}_2\mathbf{F}_1\left(\begin{matrix} a,b \\ c \end{matrix}; x\right) &= (1-x)^{c-a-b} {}_2\mathbf{F}_1\left(\begin{matrix} c-a, c-b \\ c \end{matrix}; x\right), \\
{}_2\mathbf{F}_1\left(\begin{matrix} a,b \\ c-1 \end{matrix}; x\right) &= \sum_{m=0}^{\infty} \frac{(a)_m (b)_m}{(c-1)_{2m}} x^m {}_2\mathbf{F}_1\left(\begin{matrix} a+m, b+m \\ c+2m \end{matrix}; x\right).
\end{aligned}
\tag{A.1}
$$

- The Appell hypergeometric function $\mathbf{F}_1$ has the analytic continuation as follows,

$$
\begin{aligned}
\mathbf{F}_1\left(a, b_1, b_2, c, x, y\right) = & \frac{\Gamma(c)\Gamma(c-a-b_1-b_2)}{\Gamma(c-a)\Gamma(c-b_1-b_2)} \\
& \times \mathbf{F}_1\left(a, b_1, b_2, 1+a+b_1+b_2-c, 1-x, 1-y\right) \\
& + \frac{\Gamma(c)\Gamma(a+b_2-c)}{\Gamma(a)\Gamma(b_2)}(1-x)^{-b_1}(1-y)^{c-a-b_2} \\
& \times \mathbf{F}_1\left(c-a, b_1, c-b_1-b_2, c-a-b_2+1, \frac{1-y}{1-x}, 1-y\right) \\
& + \frac{\Gamma(c)\Gamma(c-a-b_2)\Gamma(a+b_1+b_2-c)}{\Gamma(a)\Gamma(b_1)\Gamma(c-a)}(1-x)^{c-a-b_1-b_2} \\
& \times G_2\left(c-b_1-b_2, b_2, a+b_1+b_2-c, c-a-b_2, x-1, \frac{1-y}{x-1}\right),
\end{aligned}
\tag{A.2}
$$

and

$$
\begin{aligned}
\mathbf{F}_1\left(a, b_1, b_2, c, x, y\right) = & \frac{\Gamma(c)\Gamma(a-b_1-b_2)}{\Gamma(a)\Gamma(c-b_1-b_2)}(-x)^{-b_1}(-y)^{-b_2} \\
& \times \mathbf{F}_1\left(1+b_1+b_2-c, b_1, b_2, 1+b_1+b_2-a, \frac{1}{x}, \frac{1}{y}\right) \\
& + \frac{\Gamma(c)\Gamma(b_2-a)}{\Gamma(b_2)\Gamma(c-a)}(-y)^{-a}\mathbf{F}_1\left(a, b_1, 1+a-c, 1+a-b_2, \frac{x}{y}, \frac{1}{y}\right) \\
& + \frac{\Gamma(c)\Gamma(a-b_2)\Gamma(b_1+b_2-a)}{\Gamma(a)\Gamma(b_1)\Gamma(c-a)}(-x)^{b_2-a}(-y)^{-b_2} \\
& \times G_2\left(1+a-c, b_2, b_1+b_2-a, a-b_2, -\frac{1}{x}, -\frac{x}{y}\right).
\end{aligned}
\tag{A.3}
$$

- **Evaluating the delta function:** The momentum conserving delta functions are always of much importance here because they put strong constraints on the celestial correlators. For non-vanishing values of $\alpha$ and $\beta$ we get the tree amplitude as

$$
\mathbb{M}_4^{\text{tree}}(s, t) \rightarrow -\frac{1}{48} s\left(\frac{(3\kappa + \alpha s - 8\beta s)}{(\kappa - \alpha s)(\kappa - 2\beta s)} - \frac{4}{\kappa + \alpha s} + \frac{1}{\kappa + 2\beta s} + \frac{24s}{t\kappa}\right) + \mathcal{O}\left(\frac{t}{s}\right) + \ldots \tag{A.4}
$$

The momentum conserving delta function can also be written in terms of the simplex variables [4] as $\sigma_i = v^{-1}\omega_i$ with $\sum_{i=1}^n \sigma_i = 1$,

$$
\begin{aligned}
\prod_{i=1}^n \int_0^\infty d\omega_i \, \omega_i^{i\lambda_i}[\cdots] = & \int_0^\infty dv \, v^{\sum i\lambda_i - 1} \prod_{i=1}^n \int_0^1 d\sigma_i \sigma_i^{i\lambda_i} \\
& \times \delta^{(4)}\left(\sum_{i=1}^4 \epsilon_i \sigma_i q_i\right)\delta\left(\sum_{i=1}^4 \sigma_i - 1\right)[\cdots].
\end{aligned}
\tag{A.5}
$$

We can cast the delta function as [4]

$$
\begin{aligned}
\delta^{(4)}\left(\sum_{i=1}^4 \epsilon_i \sigma_i q_i\right)\delta\left(\sum_{i=1}^4 \sigma_i - 1\right) = & \frac{1}{4}\delta(|z_{12}z_{34}\bar{z}_{13}\bar{z}_{24} - z_{13}z_{24}\bar{z}_{12}\bar{z}_{34}|) \\
& \times \delta\left(\sigma_1 + \frac{\epsilon_1\epsilon_4}{\mathcal{D}_4}\frac{z_{24}\bar{z}_{34}}{z_{12}\bar{z}_{13}}\right)\delta\left(\sigma_2 - \frac{\epsilon_2\epsilon_4}{\mathcal{D}_4}\frac{z_{34}\bar{z}_{14}}{z_{23}\bar{z}_{12}}\right) \\
& \times \delta\left(\sigma_3 + \frac{\epsilon_3\epsilon_4}{\mathcal{D}_4}\frac{z_{24}\bar{z}_{14}}{z_{23}\bar{z}_{13}}\right)\delta\left(\sigma_4 - \frac{1}{\mathcal{D}_4}\right) \\
\equiv & \frac{1}{4}\delta(|z_{12}z_{34}\bar{z}_{13}\bar{z}_{24} - z_{13}z_{24}\bar{z}_{12}\bar{z}_{34}|)\prod_{i=1}^4 \delta\left(\sigma_i - \sigma_{\star i}\right),
\end{aligned}
\tag{A.6}
$$

where the denominator $\mathcal{D}_4$ is defined as

$$\mathcal{D}_4 = \left(1 - \epsilon_1 \epsilon_4\right) \frac{z_{24}\bar{z}_{34}}{z_{12}\bar{z}_{13}} + \left(\epsilon_2 \epsilon_4 - 1\right) \frac{z_{34}\bar{z}_{14}}{z_{23}\bar{z}_{12}} + \left(1 - \epsilon_3 \epsilon_4\right) \frac{z_{24}\bar{z}_{14}}{z_{23}\bar{z}_{13}}. \tag{A.7}$$

In the above equation, $\sigma_{*i}$ are the supports of the $\sigma$ integral, using localized Dirac delta functions. On the support of the delta functions, the Mandelstam variables simplify to $s = v^2, t = -zv^2$. This parametrization is valid for massless particles as external legs. Hence the $\sigma_i$ integral becomes

$$\int_0^1 d\sigma_i \sigma_i^{i\lambda_i} \delta\left(\sum_i \sigma_i - 1\right) \delta^{(4)}\left(\sum_{i=1}^4 \epsilon_i \sigma_i q_i\right)$$

$$= \frac{1}{4}\delta(|z_{12}z_{34}\bar{z}_{13}\bar{z}_{24} - z_{13}z_{24}\bar{z}_{12}\bar{z}_{34}|) \prod_{i=1}^4 \int_0^1 d\sigma_i \sigma_i^{i\lambda_i} \delta\left(\sigma_i - \sigma_{*i}\right) \tag{A.8}$$

$$\sim (z-1)^{\frac{\Delta_1 - \Delta_2 - \Delta_3 + \Delta_4}{2}} |z|^{-\Delta_1 - \Delta_2} \frac{\delta(|z-\bar{z}|)}{|z_{13}|^2|z_{24}|^2} \prod_{i=1}^4 \underbrace{\mathbf{1}_{[0,1]}(\sigma_{*i})}_{\text{indicator function}},$$

where, the indicator function ensures all the $\sigma_{i*}$ are between 0 to 1.

$$\mathbf{1}_{[0,1]}(\sigma_{*i}) = \begin{cases} 1, & \text{if } \sigma_{*i} \in [0,1], \\ 0, & \text{otherwise.} \end{cases} \tag{A.9}$$

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
