# Peer review of "The Sky Remembers everything: Celestial amplitude, Shadow and OPE in quadratic EFT of gravity"

_SciPost Physics, doi:SciPost Phys. 19, 041 (2025)_

## Round 1 · Referee Report · Anonymous (Referee 1) · 2025-6-11

Report
The authors compute celestial amplitudes involving higher curvature corrections to Einstein gravity. In particular, they consider quadratic gravity shown in Eq.(3.1). As they summarize in the introduction and conclusion, the main new results are
1) Eikonal amplitudes for the quadratic EFT of gravity
2) UV and IR behaviour with the EFT corrections
3) Celestial Born amplitude with the EFT corrections and the shadow corrector associated with it. They also showed two methods of extracting the celestial OPEs
These are interesting new results of celestial amplitudes presented in a well-written manuscript.
However, as the authors have pointed out, their work is highly motivated and based on existing work in the literature. To meet the standard of Sci Post, I am willing to reconsider this paper for publication, provided the following comments and questions are addressed.
Requested changes
1) In the third paragraph of the introduction, Ref.[30] was not the first example of computing shadow correlators for celestial amplitude. Actually Ref.[20] provided the first example for four-point. I invite the authors revise the references for that sentence.
2) Above Eq.(2.3), the expression for the holomorphic conformal weight holds for the scalar case but not for the spinning case as the authors explain in Eq.(2.6). I would recommend the authors to revise the sentences above Eq.(2.3).
3) The quadratic gravity is introduced from section 3. It might be useful to discuss a bit more about the basic properties of the quadratic gravity. Including some useful references might be helpful. For example, it might be helpful to explain how the mass of the massive mode of it is related to the parameters in Eq.(3.1). And is it clear that the amplitudes computed in quadratic gravity are unitary?
4) The sentence below Eq.(3.11) is confusing. From Eq.(3.10), the Einstein gravity term is still present with the distributional nature. The EFT correction term behaves better but the entire celestial amplitude is not improved by that.
5) In Eq.(3.15), the authors computed the eikonal phase in quadratic gravity by perform Fourier transform on the modified Born amplitudes, following the same procedure in GR. It might be useful to explain why in the presence of the EFT corrections, the eikonal phase can still be computed in the same way as GR. Or are there other similar examples that existed in the literature can justify this point.
6) Although the authors presented a nice discussion on the delta function on page 11, it is not very clear how the author obtained the (e-1) factor in Eq.(3.22). I hope the authors could explain a bit more below Eq.(3.22).
7) In Eq.(4.11), the authors did not thow the contribution from GR. I am wondering if the authors had tried to compute it. If it is not doable for general conformal dimensions, one might try to take some specific limit of the conformal dimension. See, e.g. 2501.05805.
8) In Eq.(5.1), the physical meaning of n in the second line seems to be related to the number of particles. It might be helpful to clarify it.
I hope the comments and questions above might help the authors improve the manuscript.
Recommendation
Ask for minor revision
Dear Editor,
Please see the attached file for point-wise response to the Referee report.
Best,
Authors
Attachment:
I thank the authors for the response. The authors have addressed the questions and concerns I had previously in their response. I would be happy to recommend the revised version for publications.

Author: Saptaswa Ghosh on 2025-07-10 [id 5628]
(in reply to Report 2 on 2025-07-09)Dear Editor,
Please see the attached file for point-wise response to the Referee report.
Best,
Authors
Attachment:
response_ref2.pdf

---

## Round 1 · Referee Report · Anonymous (Referee 2) · 2025-7-9

Report
Recommendation
Ask for major revision

---

## Round 2 · Author Response

We are resubmitting the revised version. The response to the queries raised by the referees has been addressed in individual authors' comment. The associated changes are colour coded (blue for the first Referee and violet for the second Referee.)
Best,
Authors.

---

## Round 2 · List of Changes

First Referee:
- Explanation regarding the unitarity of quadratic EFT is added after (3.4) , page: 7-9.
2.Explanation regarding the eikonal exponentiation is added in Sec 3.2, page: 12-13.
-
Footnote (5) added explaining the analytic linearization of the exponential map.
-
Derivation of GR contribution is added in (4.12).
-
Typo fixed regarding the particle number "n" in (5.1).
Second Referee:
-
The scalar matter is included explicitly in (3.7).
-
$\gamma$ is explicitly defined after (2.10).
-
Footnote (3) added regarding notations and order of operations mentioned in pt. 4.
-
$\delta_1,2$ are defined in (3.13).
-
The relation between t and q^2 is defined (3.15).
-
Large $\omega$ limit explained after (3.35).
-
Notational change regarding the OPE coefficients made around (4.27, 4.28)

---

## Editorial Decision

published